# SSL4RL: Revisiting Self-supervised Learning as Intrinsic Reward for Visual-Language Reasoning

Xiaojun Guo [* 1 2]  Runyu Zhou [* 1]  Yifei Wang [3]  Qi Zhang [1]  Chenheng Zhang [1]  Stefanie Jegelka [4 5]
Xiaohan Wang [2]  Jiajun Chai [2]  Guojun Yin [2]  Wei Lin [2]  Yisen Wang [1]

## Abstract

Vision-language models (VLMs) have shown remarkable abilities by integrating large language models with visual inputs. However, they often rely on textual shortcuts rather than adequately using visual evidence during reasoning. Although reinforcement learning (RL) can align models with desired behaviors, its application to VLMs has been hindered by the lack of scalable and reliable rewards. To overcome this challenge, we propose **SSL4RL**, a novel framework that leverages self-supervised learning (SSL) tasks as a source of verifiable rewards for RL. Our approach reformulates SSL objectives like rotation prediction and patch reconstruction into dense automatic rewards, removing the need for human preferences or AI evaluators. Experiments show that SSL4RL substantially improves performance on both vision-centric and vision-language reasoning benchmarks, with encouraging potentials on open-ended scenarios and stronger resilience to visual corruptions. Through systematic ablations, we identify key factors influencing SSL4RL, including data volume, model scale, model choice, task combination, and task difficulty, thereby offering new design principles for future work. Our implementation is open-sourced at https://github.com/PKU-ML/SSL4RL, with models hosted on Huggingface collection PKU-ML/ssl4rl.

*Equal contribution [1]State Key Lab of General Artificial Intelligence, School of Intelligence Science and Technology, Peking University [2]Meituan [3]Amazon AGI SF Lab [4]Technical University of Munich, School of CIT, MCML, MDSI [5]Massachusetts Institute of Technology, EECS and CSAIL. Correspondence to: Yisen Wang <yisen.wang@pku.edu.cn>.

*Proceedings of the 43[rd] International Conference on Machine Learning*, Seoul, South Korea. PMLR 306, 2026. Copyright 2026 by the author(s).

## 1. Introduction

Vision–Language Models (VLMs) have rapidly advanced multimodal understanding by leveraging the expert-level reasoning capabilities of Large Language Models (LLMs). This synergy has enabled broad applications, from visual question answering to interactive dialogue. Yet the reliance on linguistic priors introduces systematic weaknesses. In *vision-centric* tasks such as classification, where answers must be derived solely from image content, VLMs often lag behind specialist vision models (Fu et al., 2025; Tong et al., 2024; Fu et al., 2024). In *vision–language reasoning* tasks, VLMs tend to exploit textual knowledge rather than grounding their reasoning in visual evidence, a tendency amplified in long-form generation (Jian et al., 2025). These limitations underscore the need for training methods that reinforce visual grounding and robust reasoning simultaneously.

Reinforcement Learning (RL) has emerged as the dominant paradigm for post-training large models, demonstrating that preference-based signals derived from either humans or AI feedback can substantially improve helpfulness and alignment (Li et al., 2025b; Ouyang et al., 2022; Rafailov et al., 2023; Bai et al., 2022; Li et al., 2025a). More recently, *verifier-driven RL* has shown particular promise: models trained with automatically checkable rewards achieve striking gains in domains such as mathematics and programming (Le et al., 2022; Shao et al., 2024). However, these successes expose a fundamental bottleneck: outside domains with explicit programmatic verifiers, scalable and reliable rewards are scarce. In such cases, training pipelines often revert to *LLM-as-a-judge* heuristics, which are biased, noisy, and prone to adversarial manipulation (Raina et al., 2024; Chen et al., 2024b). This raises a key question:

> *How can we obtain abundant, verifiable reinforcement signals for VLMs in domains where external verifiers are absent?*

Self-supervised learning (SSL) offers a natural but underexplored answer. SSL has been central to representation learning across modalities: masked language modeling and next-token prediction for text (Devlin et al., 2019; Lewis et al., 2020; Zhang et al., 2024a), contrastive learning and

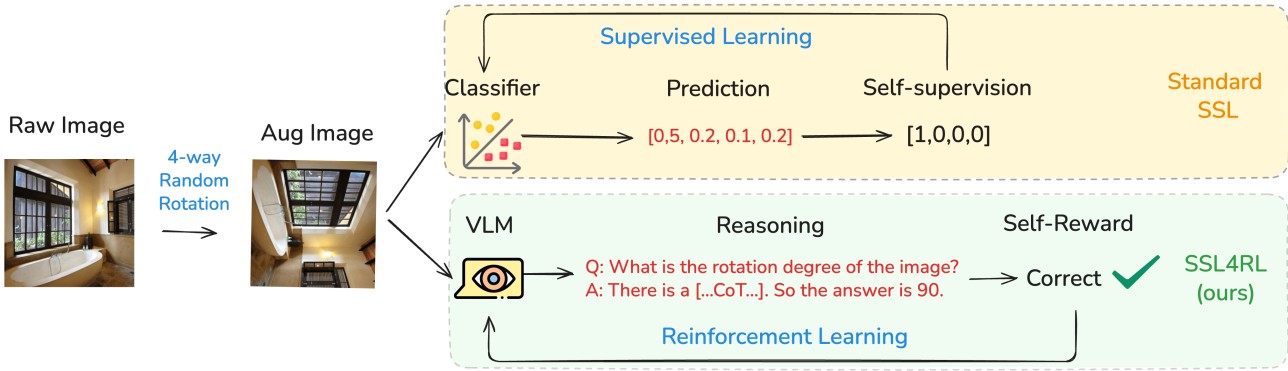

Figure 1. Overview of the SSL4RL framework. A corruption function transforms an input into a context–target pair. The model conditions on the context, generates predictions, and receives a verifiable reward by comparing against the target. The reward is then used to optimize the model via Reinforcement Learning (RL).

masked autoencoders for vision (Zhang et al., 2025; Chen et al., 2020; Grill et al., 2020; Caron et al., 2021; He et al., 2022; Zhang et al., 2022). The principle is simple yet powerful: perturb data, and require the model to reconstruct or discriminate. Crucially, SSL tasks define *intrinsically verifiable targets*. Given an image and its rotated variant, the ground-truth angle is unambiguous. Unlike preference-based signals, these targets are properties of the data itself, providing dense, reliable supervision at scale.

In this paper, we propose **SSL4RL**, a general framework that repurposes SSL tasks as verifiable reward functions for RL-based post-training. Instead of treating SSL solely as a pre-training tool (Tong et al., 2024), we reinterpret it through the lens of RL: corrupted inputs define trajectories, correctness defines rewards, and policy optimization drives updates. SSL4RL requires no human labels, external verifiers, or heuristic judges, yet produces dense and scalable reinforcement signals. Importantly, unlike conventional SSL, SSL4RL emphasizes generating natural language reasoning paths to solve vision tasks, thereby bridging perceptual learning and reasoning alignment.

We present a comprehensive evaluation of SSL4RL spanning a wide range of tasks. In vision-centric perception, SSL4RL significantly improves ImageNet-1K classification accuracy over the base model. For vision–language reasoning, it delivers consistent gains with average improvements of 7.03%. In the open-ended image captioning scenario, SSL4RL surpasses the base model by up to 8.14 points. A key finding is that the effectiveness of SSL4RL tasks differs from their role in traditional SSL: contrastive objectives, while dominant in pre-training, show limited benefit unless paired with stronger data augmentation, whereas position prediction, which is often deemed too trivial for SSL, emerges as highly effective in the SSL4RL setting. Our ablation study reveals that SSL4RL is robust to base model choice, scales favorably with data, but is sensitive to task

difficulty. These findings offer initial insights into what makes a good SSL4RL task.

**Contributions.** Our work makes three key contributions:

- We introduce SSL4RL, a unified framework bridging self-supervised learning and RL-based post-training through verifiable rewards.

- We provide a comprehensive study identifying which SSL tasks best transfer to reasoning and which do not.

- We conduct comprehensive ablation studies on model capacity, base model choice, data volume scaling, difficulty scaling, and task combinations.

Together, these results emphasize that the abundance of verifiable signals in self-supervised learning can be harnessed not only for representation learning, but also to drive alignment and reasoning in VLMs.

## 2. Related Work

**RL training with external verifiers.** Reinforcement learning has become a dominant paradigm for aligning large models. RLHF aligns LLMs with human intent through preference data (Ouyang et al., 2022), while Direct Preference Optimization (DPO) reframes preference learning as a contrastive loss without explicit reward models (Rafailov et al., 2023). Constitutional AI replaces human raters with rule-based AI feedback (Bai et al., 2022). More recently, *verifier-driven RL* has achieved notable success in domains such as code and math, where correctness can be mechanically checked (Le et al., 2022; Shao et al., 2024). However, in domains lacking such verifiers, many systems fall back on *LLM-as-a-judge* signals. While convenient, these rewards are biased, noisy, and adversarially manipulable (Raina et al., 2024; Chen et al., 2024b), undermining their

reliability. This limitation motivates the search for scalable alternatives where correctness is intrinsic to the data.

**RL training with self-reward.** Several methods attempt to reduce dependence on external labels or verifiers by enabling models to generate their own training signals. Self-Instruct (Wang et al., 2023b) bootstraps fine-tuning data through synthetic instructions, while STaR (Zelikman et al., 2022) and Reflexion (Shinn et al., 2023) improve reasoning via self-generated rationales and feedback. Self-Consistency (Wang et al., 2023a) aggregates multiple sampled rationales to increase reliability. Reinforced Pre-Training (RPT) scales this idea by using RL objectives such as next-token prediction at the pretraining stage (Dong et al., 2025). While these methods reduce the need for human labels, they still optimize toward approximating *original* task accuracy, often requiring correctness evaluation or model-judging heuristics. In contrast, our approach seeks verifiable, abundant signals outside the original task by reinterpreting SSL tasks as reinforcement rewards.

**Self-supervised learning across modalities.** Self-supervision has been the foundation of representation learning across domains. In language, pretext tasks include masked language modeling (Devlin et al., 2019; Lewis et al., 2020) and next-token prediction. Variants that mask reasoning steps show promise in enhancing mathematical reasoning (Chen et al., 2024a). In vision, early tasks include rotation (Gidaris et al., 2018), jigsaw (Noroozi & Favaro, 2016), and context prediction (Doersch et al., 2015), while modern SSL emphasizes contrastive learning (Chen et al., 2020; He et al., 2020) and generative masking (He et al., 2022; Du et al., 2024). In multimodal learning, CLIP (Radford et al., 2021) and ALBEF (Li et al., 2021) demonstrate the power of contrastive and distillation objectives for aligning vision and language (Zhang et al., 2023). A unifying property of all these objectives is their *intrinsically verifiable targets*, making them natural candidates for repurposing as RL rewards. For vision-language models, Jigsaw-R1 (Wang et al., 2025) first establishes jigsaw puzzles as an effective pretext task for reinforcement learning. In comparison, we go beyond one specific task and generalize the concept into a unified SSL4RL framework. Through comprehensive studies, we systematically investigate the scaling of data volume, model size, task difficulty, and task combinations, offering broader insights into this learning paradigm.

## 3. SSL4RL Framework

We formalize SSL4RL as a general recipe for converting self-supervised learning (SSL) objectives into reinforcement learning (RL) rewards for post-training large language models and related architectures. This section introduces the notation, shows how SSL tasks are reinterpreted under the RL formalism, and describes the optimization strategy. A high-level illustration of the framework is shown in Figure 1.

**Problem Setup**. Let $\pi_\theta$ denote a parametric model with parameters $\theta$, defined over sequences of actions. In vision–language tasks, actions may include discrete classification labels (*e.g.*, rotations, patch indices) or text tokens. We assume access to a data distribution $\mathcal{D}$ of inputs $x \in \mathcal{X}$, which can be text and images. In the standard RL formalism, a trajectory $\tau$ is generated by rolling out $\pi_\theta$ in an environment $\mathcal{E}$, and receives a scalar reward $R(\tau)$. In SSL4RL, the "environment" is defined by a corruption function $c(x) = (\tilde{x}, y)$, which maps an input $x$ into a corrupted context $\tilde{x}$ and a ground-truth target $y$. The policy $\pi_\theta$ conditions on $\tilde{x}$ to produce an output $\hat{y}$, and a reward $r(\hat{y}, y)$ is computed based on agreement with the ground truth. Thus, every SSL task induces an RL task: the corruption defines the environment, the target defines the verifiable ground truth, and the reward function provides supervision.

**From SSL to RL Rewards.** Our work revisits four representative SSL pretext tasks, depicted in Figure 2. Each SSL task is defined by a tuple $(c, y, r)$, where $c$ is a corruption function applied to the inputs $x$ (specifically referring to images), $y$ is the self-supervised target derived from the corruption, and $r$ is the reward signal computed based on the model's prediction $\hat{y}$. Specifically,

- ***Rotation Prediction*** (Gidaris et al., 2018): $c(x, y)$ rotates $x$ by an angle $y$. The reward is $r = \mathbb{1}[\hat{y} = y]$, where $\hat{y}$ is the predicted angle.

- ***Jigsaw Puzzles*** (Noroozi & Favaro, 2016): $c(x, y)$ partitions $x$ into a grid and permutes patches by index $y$. The reward is $r = \mathbb{1}[\hat{y} = y]$, where $\hat{y}$ is the predicted permutation.

- ***Contrastive Learning*** (Chen et al., 2020; Radford et al., 2021): $c(x)$ generates augmented views. The reward $r$ is a binary classification, imitating the InfoNCE similarity score that encourages high similarity for positive pairs and low similarity for negatives.

- ***Patch Position Prediction*** (Doersch et al., 2015): $c(x, y)$ extracts a patch from location $y$. The reward is $r = \mathbb{1}[\hat{y} = y]$, where $\hat{y}$ is the predicted location.

These rewards are *verifiable* as they are computed against unambiguous ground-truth targets $y$.

**Optimization via GRPO**. We adopt Grouped Reinforcement Policy Optimization (GRPO) (Shao et al., 2024), an efficient policy-gradient method designed for large-scale LLM training. Given a reference policy $\pi_0$ (e.g., the supervised fine-tuned model before RL), GRPO optimizes $\pi_\theta$ by maximizing the following regularized objective:

$$\mathcal{J}(\theta) = \mathbb{E}_{\tau \sim \pi_\theta}[R(\tau) - \beta \, \mathrm{KL}(\pi_\theta(\cdot|\tau) \,\|\, \pi_0(\cdot|\tau))], \quad (1)$$

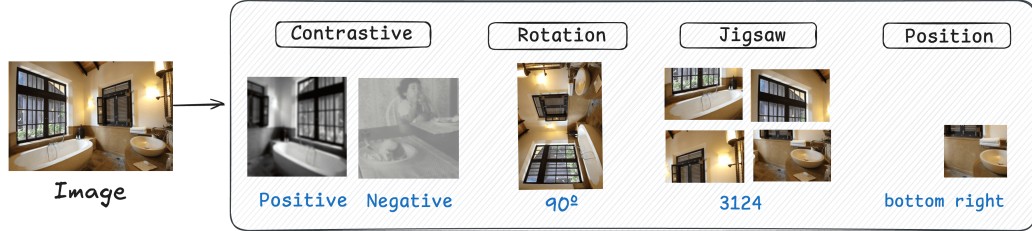

*Figure 2.* Four SSL4RL tasks considered in our study. *Rotation*: An image is rotated by a predefined angle, and the task is to predict this angle. *Jigsaw*: After dividing an image into a grid and permuting the patches, the goal is to predict the correct permutation index. *Contrastive*: Two augmented views are generated from an image, and the objective is to identify whether two given views originate from the same source image. *Position*: Given an image and a single patch cropped from it, the task is to predict the patch's original position.

where $R(\tau)$ is the SSL-derived reward, and $\beta$ controls the strength of KL regularization. The KL penalty prevents divergence from the reference distribution. GRPO performs updates by sampling rollouts, normalizing rewards across groups, and applying clipped policy-gradient updates similar to PPO (Schulman et al., 2017), but in a manner more compute-efficient for large batch LLM training. All experiments in this work apply the same GRPO configuration across tasks, ensuring that differences in outcomes are attributable to the choice of SSL reward.

## 4. What Makes a Good SSL4RL Task?

In this section, we examine the core principles for designing effective SSL4RL tasks through comprehensive experiments on vision-language reasoning (Section 4.1.1), vision-centric benchmarks (Section 4.1.2), and open-ended image-captioning (Section 4.1.3). We propose a stage-wise task combination strategy in Section 4.2 and provide a robustness analysis in Section 4.3. In Section 4.4, we present ablation studies on data volume, base model, and task difficulty to elucidate these design principles further.

### 4.1. Experiments on Downstream Vision Tasks

#### 4.1.1. VISION-LANGUAGE REASONING TASKS

**SSL Task Settings.** For each SSL pretext task, we implement the following configurations. In Rotation, images are rotated counterclockwise by a randomly selected angle from 0°, 90°, 180°, 270°. In Jigsaw, each image is partitioned into a 2×2 grid, and the patches are randomly shuffled. In Contrastive, we apply the standard augmentation pipeline from Chen et al. (2020), including color jittering, grayscale conversion, Gaussian blur, horizontal flipping, and random resized cropping, with a probability of 0.2. In Position, the image is divided into four equal quadrants, and the target is to identify which quadrant contains a specified patch.

**Training and Evaluation Settings.** Our experiments primarily adopt Qwen-2.5-VL-3B/7B-Instruct (Bai et al., 2025) for their moderate sizes and strong reasoning performance.

For GRPO training, we set the group size to be 5, the KL loss coefficient to be 0.01, the entropy loss coefficient to be 0, and the context length to be 2048. We train the models on 8 A800 GPUs with a batch size of 512. For evaluation, we adopt the third-party evaluation tool VLMEvalKit (Duan et al., 2024), with the default sampling configured with a temperature of 0.01, top-p of 0.001, and top-k of 1. We visualize the rewards, entropy, and response length trajectories during RL training in Appendix F.

**Benchmarks.** We assess our approach on six prominent Vision Questioning Answer (VQA) benchmarks: MMBench (Liu et al., 2024), SEED-Bench (Li et al., 2023), V* (Wu & Xie, 2024), RealWorldQA (xAI, 2024), BLINK (Fu et al., 2024), and MME-RealWorld (Zhang et al., 2024b), featuring challenging VLMs' ability on visual perception, spatial understanding, detail capturing, and so on. Detailed datasets descriptions are provided in Appendix E.1.

**Results.** Overall, the proposed SSL4RL paradigm leads to substantial performance gains on vision questioning answer tasks. Shown in Table 1, on average, SSL4RL models outperform the base model by 7.03% across benchmarks, especially on V* (+8.90%) and RealWorldQA (+9.55%). For reference, we provide the leaf task results of MMBench and SEED-Bench in Appendix A. These consistent improvements validate the effectiveness of SSL4RL in enhancing vision-language reasoning. To contextualize the performance of our SSL4RL method, we include a strong baseline *VLM-R1* (Shen et al., 2025). We fine-tune the base model on 60% of the benchmark and evaluate on the held-out test set. The RL reward is computed directly based on the *golden* answer. We denote the tuned model as **Golden-3B** (see more details in Appendix D). As shown in Table 2 and 3, the performance gap between our SSL4RL variants and the Golden-3B oracle is relatively small, *e.g.*, 81.35% vs. 84.93% on MMBench, and 69.80% vs 73.21% on SEED-Bench, compared to our improvements over base models. This demonstrates that our self-supervised objectives, which require *no labeled downstream data*, can effectively close the gap to the performance driven by idealized, task-specific reward signals.

*Table 1.* Test performance (%) on VQA benchmarks.

| Category | Model | MMBench | SEED-Bench | V* | RealWorldQA | BLINK | MME | *Average* |
|---|---|---|---|---|---|---|---|---|
| Base | Qwen2.5-VL-3B | 72.99 | 60.83 | 59.16 | 52.67 | 42.13 | 32.41 | 53.36 |
| SSL4RL-3B | Rotation | **80.38** | 69.10 | 62.30 | 62.09 | **48.18** | 34.18 | 59.37 |
| | Jigsaw | 77.82 | 67.67 | 63.35 | **62.22** | 45.18 | 35.12 | 58.56 |
| | Contrastive | 69.27 | 61.90 | 60.20 | 58.03 | 45.13 | 30.17 | 54.11 |
| | Position | 80.08 | **69.77** | **68.06** | 59.86 | 46.39 | **38.19** | **60.39** |
| *Maximal Improvement* | | ↑ **7.39** | ↑ **8.94** | ↑ **8.90** | ↑ **9.55** | ↑ **6.05** | ↑ **5.78** | ↑ **7.03** |

*Table 2.* Performance (%) of on the MMBench test set. Logical: Logical Reasoning, Relation: Relation Reasoning, Attribute: Attribute Reasoning, Coarse: Coarse Perception, Cross Inst.: Cross-Instance Fine-grained Perception, Single-Inst.: Single-Instance Fine-grained Perception.

| Category | Model | Logical | Relation | Attribute | Coarse | Cross-Inst. | Single-Inst. | *Average* |
|---|---|---|---|---|---|---|---|---|
| Base model | Qwen2.5-VL-3B | 53.03 | 46.19 | 77.10 | 74.56 | 67.94 | 81.41 | 72.63 |
| RL with golden rewards | Golden-3B | 71.82 | 86.70 | 88.34 | 83.74 | 79.57 | 89.39 | 84.93 |
| SSL4RL-3B | Rotation | 66.06 | **82.97** | 87.38 | 80.07 | **75.07** | 85.93 | **81.35** |
| | Jigsaw | 62.53 | 79.06 | 85.84 | 77.39 | 66.43 | 84.14 | 78.46 |
| | Contrastive | 60.50 | 71.50 | 73.75 | 66.31 | 56.17 | 77.72 | 68.82 |
| | Position | **67.58** | 78.37 | 85.16 | **81.22** | 70.70 | **87.87** | 81.12 |

*Table 3.* Performance (%) on the SEED-Bench test set. TU: Text Understanding, VR: Visual Reasoning, SU: Scene Understanding, IId: Instance Identity, IIn: Instance Interaction, IA: Instance Attributes, IL: Instance Location, SR: Spatial Relation, IC: Instances Counting.

| Category | Model | TU | VR | SU | IId | IIn | IA | IL | SR | IC | *Average* |
|---|---|---|---|---|---|---|---|---|---|---|---|
| Base model | Qwen2.5-VL-3B | 59.04 | 60.44 | 67.17 | 63.83 | 61.80 | 51.28 | 57.20 | 59.09 | 59.54 | 62.00 |
| RL with golden rewards | Golden-3B | 67.68 | 68.41 | 74.76 | 77.03 | 74.69 | 82.05 | 63.84 | 77.27 | 77.10 | 73.21 |
| SSL4RL-3B | Rotation | 62.60 | 62.64 | 72.28 | 73.69 | 71.28 | 58.46 | 55.35 | **68.18** | **77.10** | 69.01 |
| | Jigsaw | 60.16 | 60.99 | 71.31 | 74.01 | 69.53 | 58.46 | **59.78** | 63.64 | 74.81 | 67.97 |
| | Contrastive | 55.08 | **64.01** | 69.79 | 68.76 | 61.59 | 30.77 | 56.09 | 56.82 | 74.05 | 63.02 |
| | Position | **63.21** | 62.36 | **72.83** | **74.17** | **72.40** | **58.97** | 57.93 | 65.91 | 73.28 | **69.80** |

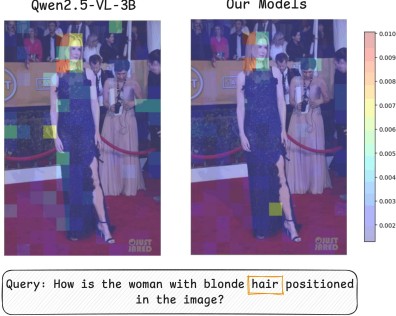

*Figure 3.* Cross-Attention Heatmap Comparison. More instances are shown in Appendix G.

*Table 4.* Test performance (%) on HallusionBench.

| Category | Model | Accuracy | *Improvement* |
|---|---|---|---|
| Base | Qwen2.5-VL-3B | 46.27 | - |
| SSL4RL | Rotation | 47.42 | ↑ **1.15** |
| | Jigsaw | 49.32 | ↑ **3.05** |
| | Contrastive | 50.26 | ↑ **3.99** |
| | Position | 52.37 | ↑ **6.10** |

**Analysis.** Our experimental analysis reveals that the Rotation and Position pretext tasks consistently yield the strongest performance gains. The success of Position is intuitive, as localizing a patch within the global image compels the model to integrate fine-grained local details with the overall scene layout, fostering integrated spatial understanding. In contrast, the remarkable effectiveness of Rotation presents a more nuanced insight. Despite its perceptual simplicity for humans, we find the task poses a considerable challenge to VLMs, as evidenced by the base model's near-chance accuracy. We interpret this as evidence that rotation prediction facilitates "anti-commonsense" learning. The model's pre-training instills a strong prior for canonical orientations, *e.g.*, people are typically depicted upright. Rotated images violate this prior, forcing the model to reconcile anomalous inputs with its existing knowledge, thereby sharpening its relational reasoning and visual comprehension. Conversely, the Contrastive task sometimes leads to performance degradation. We hypothesize that its default augmentations are insufficiently challenging, causing the model to overfit to superficial invariances without learning semantically meaningful structures. Subsequent ablation studies confirm this: employing a more aggressive aug-

mentation strategy recovers significant performance. This finding highlights the critical need to align SSL task difficulty with model capacity to elicit generalizable feature learning, as further explored in Section 4.4.2.

**Observations.** Through a qualitative analysis of model responses, we observe two key improvements attributable to SSL4RL training. (1) **Sharper Attention**: The trained models exhibit more precise attention alignment with text queries. For instance, when queried about "hair" (Figure 3), the base model's attention is diffuse, whereas our model accurately focuses on the relevant region. This is not merely an effect of sparsity, as evidenced by the "sky" query (Appendix Figure 12): our model attends to the entire sky, while the base model activates only scattered pixels. (2) **Reduced Language Bias**: SSL4RL mitigates over-reliance on linguistic priors, fostering greater dependence on visual evidence. For example, when asked about a chandelier's color (Appendix Figure 11), the base model defaults to a common-sense response (*e.g.*, a typical decorative color), while our model localizes the object and answers based on

the actual appearance. We also evaluate on HallusionBench (Guan et al., 2024), a test for overriding linguistic priors that conflict with visual input. As shown in Table 4, SSL4RL models consistently improve, with Position gaining +6.10%. This offers quantitative evidence of reduced language bias and better visual grounding.

### 4.1.2. VISION-CENTRIC TASKS: IMAGENET CLASSIFICATION

**Benchmarks.** We further evaluate the SSL4RL paradigm on vision-centric tasks using the ImageNet-1K dataset (Deng et al., 2009), which comprises approximately 1.3 million images across 1,000 categories. From this dataset, we construct a balanced subset of 100,000 training and 10,000 test images. To probe reasoning capabilities at varying difficulty levels, we design three question types: (1) *Completion*: directly answer with the species name; (2) *Choice-20*: select the correct species from 20 candidates; (3) *Choice-200*: select the correct species from 200 candidates.

**Results.** As shown in Table 5, models fine-tuned with SSL4RL consistently outperform the base model across all question types on the ImageNet-1K classification task. Consistent with findings on reasoning benchmarks, the Position task leads to the largest performance gains, *e.g.*, 67.14% vs. 57.20% on Choice-200. However, a key divergence emerges with the Contrastive task. While it underperformed on vision-language reasoning, it shows competitive results on ImageNet classification. We attribute this result to the nature of the downstream benchmark. As an instance discrimination task, ImageNet benefits from learning strong semantic representations through invariance to augmentations—precisely the strength of contrastive learning. This result also verifies that SSL4RL task selection must consider the specific capabilities required by the target application.

*Table 5.* Test performance (%) on ImageNet downstream tasks.

| Category | Model | Completion | Choice-20 | Choice-200 |
|---|---|---|---|---|
| Base Model | Qwen2.5-VL-3B | 24.93 | 85.22 | 57.10 |
| SSL4RL | Rotation | **29.19** | 87.26 | 58.48 |
| | Jigsaw | 28.75 | 87.55 | 60.80 |
| | Contrastive | 26.84 | 89.51 | 61.78 |
| | Position | 28.76 | **92.35** | **67.14** |

### 4.1.3. OPEN-ENDED VISION-LANGUAGE TASK: IMAGE CAPTIONING

SSL4RL requires no human labels, external verifiers, or heuristic judges, yet produces dense and scalable reinforcement signals. Intuitively, it has potential for open-ended tasks where ground truth is ill-defined and human annotations are expensive.

**Benchmarks and Results.** To evaluate SSL4RL's potential on open-ended tasks, we leverage a recent image captioning platform, CapArena (Cheng et al., 2025), where captions from a test model are compared against those from strong baseline models using GPT-4o as a judge, with human references provided for context. See Appendix E.2 for more details. As shown in Table 6, SSL4RL consistently improves over the base model, with the largest performance gain of 8.14% (56.45% vs. 48.31%). This demonstrates that the self-supervised rewards provide a meaningful learning signal even in the absence of a verifiable ground truth.

**Qualitative Insights.** We conduct a qualitative comparison with examples in Appendix Table 25 to illustrate the mechanisms by which SSL4RL enhances open-ended generation: (1) **Enhanced Detail Capture.** SSL4RL models correctly capture more details in the image. For instance, our SSL4RL model correctly identifies a scoreboard displaying "1 0 0" and an accessibility sign, whereas the baseline model frequently omits or misinterprets these elements. (2) **Improved Spatial Reasoning.** Captions generated by SSL4RL models tend to employ more precise spatial descriptors, such as "*behind the fence*," "*to the left of*," and "*the center of the court*," aligning well with the spatial reasoning ability required by SSL pretext tasks. Overall, these qualitative results indicate that dense self-supervised signals derived from intrinsic data structure effectively enhance model performance on the complex, open-ended task of image captioning.

*Table 6.* Scores on CapArena Platform. GPT-Score: the score against GPT-4o. Cog-Score: the score against CogVLM-19B. CPM-Score: the score against MiniCPM-8B.

| Category | Model | GPT-Score | Cog-Score | CPM-Score | *Average* |
|---|---|---|---|---|---|
| Base | Qwen2.5-VL-3B | 0.00 | 6.48 | 92.96 | 48.31 |
| SSL4RL-3B | Rotation | **19.15** | 49.30 | 96.48 | 55.28 |
| | Jigsaw | 8.87 | 54.93 | 98.59 | 55.64 |
| | Contrastive | 4.96 | 57.95 | **100.00** | **56.45** |
| | Position | 0.00 | **59.15** | 92.25 | 51.04 |

> ⚲ **Takeaway**
>
> **SSL as Intrinsic Reward Sharpens VLM Reasoning**. The SSL4RL paradigm enhances vision-language reasoning by repurposing SSL tasks as intrinsic rewards, leading to more precise attention and reduced language bias.
>
> **Task Choice is Critical**. SSL tasks should semantically align with core reasoning skills (*e.g.*, Position, Rotation) to be effective, and inappropriate tasks can cause negative transfer.
>
> **Promising for Open-Ended Tasks**. SSL4RL leads to tangible gains in tasks like image captioning, with improved spatial precision and granular detail capture.

## 4.2. Task Combination

It is natural to ask if a reliable prior exists for choosing the optimal SSL objective. However, predicting the transferability of these tasks remains an unresolved challenge (Ericsson et al., 2021; Liu et al., 2023). While specialized metrics like LogME (You et al., 2021) and RankMe (Garrido et al., 2023) have been proposed, none have reached a consensus for universal application. To bypass the pitfalls of suboptimal selection, our study shifts focus toward investigating whether a combination of multiple SSL tasks can offer a more robust solution that systematically outperforms standalone tasks.

**Naive Combination.** To explore this, we train the Qwen2.5-VL-3B model using a combination of all four SSL rewards. As illustrated in Figure 4, the combined approach, somewhat counterintuitively, does not yield significant improvements over the best single-reward setups. We hypothesize that this lack of additive improvement stems from several potential factors. First, simultaneously optimizing for multiple, distinct objectives could create a difficult optimization landscape where the model struggles to find a unified representation that satisfies all constraints, leading to interference rather than synergy. Second, the individual reward signals might vary in dynamics, making it non-trivial to balance their contributions effectively without careful reward shaping. A simple averaging of rewards might drown out the most useful learning signals. This finding suggests that a naive combination of SSL rewards is insufficient for achieving cumulative gains. It points to the need for more sophisticated integration strategies, such as dynamic reward weighting, curriculum learning that schedules different tasks, or even a meta-learner that selects the most beneficial task at different training stages.

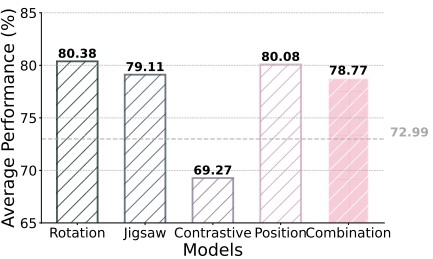

*Figure 4.* Naive combination of SSL4RL tasks on MMBench.

**Trials on Stage-wise Combination.** To avoid the instability in naive combination, we make a preliminary attempt at a stage-wise scheduling method. Unlike simple task mixing, which often creates conflicting update signals by optimizing heterogeneous objectives simultaneously, our design follows a gradual difficulty progression: Contrastive first, then Position and Rotation, with Jigsaw last. This stepwise ordering lets the model build robust global representations early, then progressively integrate spatial and compositional rea-

soning, which alleviates gradient competition and stabilizes training. As shown in Table 7, this stage-wise approach surpasses every individual SSL task, reaching 82.14% versus 80.38% on 3B models, while also delivering more reliable convergence.

*Table 7.* Stage-wise combination of SSL4RL tasks on MMBench.

| Recipe | 3B Models | 7B Models |
|---|---|---|
| Best Single Task | 80.38 | 87.73 |
| Naive Combination | 78.77 | 85.78 |
| Stage-wise Combination | **82.14** | **89.26** |

## 4.3. Robustness Analysis

SSL4RL incentivizes the model to leverage intrinsic visual structures to solve pretext tasks, rather than overfitting to superficial patterns or linguistic shortcuts. Consequently, we hypothesize that models trained with SSL4RL exhibit greater resilience to corrupted visual inputs.

**Evaluation on Perturbed MMBench.** To evaluate our model's robustness, we perturb the images in MMBench with various image transformations, including color jittering, grayscale, Gaussian blur, and horizontal flipping. We provide detailed perturbation descriptions in Appendix C. As shown in Table 8, our method consistently exhibits superior robustness across various perturbations. For instance, our SSL4RL-Rotation model achieves an average score of 72.11%, outperforming the base model at 66.80%. This performance gap persists under stronger perturbations using a multi-crop strategy in Table 9, where our SSL4RL-Position scores 64.37% compared to the base model's 59.62%. These results confirm that our SSL4RL strategy effectively enhances the model's robustness to photometric variations.

*Table 8.* Performance (%) on perturbed MMBench.

| Category | Model | Logical | Relation | Attribute | Coarse | Cross | Single | *Average* |
|---|---|---|---|---|---|---|---|---|
| Base model | Qwen2.5-VL-3B | 49.58 | 45.06 | 76.94 | 67.80 | 59.33 | 68.46 | 66.80 |
| SSL4RL-3B | Rotation | **61.00** | **72.87** | 78.66 | 70.18 | **62.57** | 72.89 | **72.11** |
| | Jigsaw | 55.47 | 70.80 | **79.38** | **71.23** | 56.28 | 71.84 | 70.84 |
| | Contrastive | 54.46 | 62.05 | 67.33 | 55.80 | 47.64 | 64.02 | 59.94 |
| | Position | 57.97 | 72.77 | 79.01 | 70.40 | 57.55 | **73.39** | 71.70 |

*Table 9.* Performance (%) on multi-crop perturbed MMBench.

| Category | Model | Logical | Relation | Attribute | Coarse | Cross-Inst. | Single-Inst. | *Average* |
|---|---|---|---|---|---|---|---|---|
| Base model | Qwen2.5-VL-3B | 45.08 | 44.99 | 71.65 | 59.88 | 53.89 | 58.62 | 59.62 |
| SSL4RL-3B | Rotation | 52.69 | 60.47 | 70.96 | 64.33 | **55.04** | 62.51 | 63.38 |
| | Jigsaw | 46.45 | 59.62 | 72.86 | 63.90 | 52.40 | 62.47 | 62.00 |
| | Contrastive | 45.56 | 46.36 | 60.86 | 49.50 | 42.14 | 53.20 | 51.25 |
| | Position | **53.43** | **61.37** | **72.95** | **65.85** | 53.68 | **65.12** | **64.37** |

**Evaluation on VLM Robustness Benchmarks.** To further investigate the resilience of SSL4RL to visual perturbations, we evaluate our models on a comprehensive VLM robustness benchmark (Ishmam et al., 2025). The benchmark comprises 213,000 synthetically corrupted images together

with 16,000 corresponding question-answer pairs. A suite of distinct corruption functions (*e.g.*, motion blur, snow, brightness transforms) is applied at five discrete severity levels. As shown in Table 10, our SSL4RL models outperform the base model across nearly all corruption functions and severity levels. Notably, the performance advantage becomes increasingly pronounced at higher corruption levels. For instance, our models surpass the base model by 5.37% on Contrast-Level5 (34.01% vs. 28.64%) and by 3.46% on Elastic-Level5 (37.16% vs. 33.70%).

*Table 10.* Performance (%) on the VLM robustness benchmark Ishmam et al. (2025). We show some corruption functions and full results can be found in Appendix Table 24.

| Level | L1 | L2 | L3 | L4 | L5 | L1 | L2 | L3 | L4 | L5 |
|---|---|---|---|---|---|---|---|---|---|---|
| | | | *Brightness* | | | | | *Contrastive* | | |
| Qwen2.5-VL-3B | 68.47 | 66.79 | 65.46 | 63.55 | 61.81 | 67.97 | 66.30 | 62.12 | 48.87 | 28.64 |
| SSL4RL-Position | 68.54 | 67.24 | 65.47 | 63.74 | 61.86 | 68.12 | 66.70 | 63.34 | 52.51 | 34.01 |
| | | | *Defocus-Blur* | | | | | *Elastic* | | |
| Qwen2.5-VL-3B | 60.25 | 57.51 | 48.11 | 40.28 | 32.95 | 65.16 | 59.60 | 61.30 | 55.72 | 33.70 |
| SSL4RL-Position | 61.35 | 59.06 | 51.22 | 44.16 | 37.25 | 65.74 | 61.13 | 62.57 | 57.21 | 37.16 |

## 4.4. Ablation Study

### 4.4.1. DATA VOLUME SCALING

A key question for data-driven methods is the ability to leverage increasing amounts of data. To characterize the data scaling for SSL4RL, we progressively expand our training dataset, obtaining three regimes of increasing data volumes: (1) **Base Set**: ~4,000 samples from MMBench; (2) **Extended Set**: ~18,000 samples from a mixture of MMBench and SEED-Bench; (3) **Full Set**: ~118,000 samples from MMBench, SEED-Bench, and ImageNet.

We select Position as a representative task for evaluation. The results, detailed in Table 11, demonstrate a clear positive scaling relationship across almost all subtasks. When scaling from Base Set to Extended Set, the average performance on MMBench improved from 80.08% to 81.38%, a gain of +1.30%. A further expansion to the Full Set yielded an additional +1.04% improvement, reaching 82.42%. This consistent, monotonic improvement suggests that the effectiveness of SSL4RL continues to benefit from larger, more diverse datasets.

*Table 11.* Performance (%) on MMBench varying data size.

| Model | Data Size | Logical | Relation | Attribute | Coarse | Cross | Single | *Average* |
|---|---|---|---|---|---|---|---|---|
| Base | – | 61.77 | 41.54 | 76.62 | 73.55 | 64.32 | 82.06 | 72.99 |
| | 4,000 | 67.65 | 77.19 | 82.22 | 82.15 | 66.51 | 85.39 | 80.08 |
| SSL4RL | 18,000 | 68.71 | 79.26 | 85.03 | 81.65 | 70.16 | 86.47 | 81.38 |
| | 118,000 | 68.77 | 79.29 | 84.26 | 83.66 | 72.87 | 88.31 | 82.42 |

### 4.4.2. BASE MODEL VARIATION

**Base Model Size.** To investigate the scalability of the SSL4RL paradigm, we apply it to the larger Qwen2.5-VL-7B-Instruct model. As shown in Table 12, our SSL4RL-7B

models surpass the base model by an average of 3.95% (69.17% vs. 65.22%), especially enhancing performance on BLINK by 8.83% and on V* by 7.86%. The relative performance ranking among the four SSL tasks is consistent with the 3B results (Table 1), with Rotation delivering the strongest gains. This consistency across model sizes verifies the robust effectiveness and promising scalability of the SSL4RL strategy.

*Table 12.* Test performance (%) on VQA benchmarks of 7B models.

| Category | Model | MMBench | SEED-Bench | V* | RealWorldQA | BLINK | MME | *Average* |
|---|---|---|---|---|---|---|---|---|
| Base | Qwen2.5-VL-7B | 86.37 | 74.70 | 73.29 | 65.88 | 45.50 | 45.59 | 65.22 |
| | Rotation | 87.50 | 75.33 | 81.15 | 68.23 | 54.33 | 48.51 | 69.17 |
| SSL4RL-7B | Jigsaw | 87.73 | 75.05 | 80.62 | 67.32 | 46.19 | 45.96 | 67.14 |
| | Contrastive | 86.25 | 75.27 | 76.96 | 66.66 | 45.60 | 45.44 | 66.03 |
| | Position | 86.25 | 75.56 | 77.48 | 67.45 | 46.76 | 47.57 | 66.84 |
| *Maximal Improvement* | | ↑ 1.36 | ↑ 0.86 | ↑ 7.86 | ↑ 2.35 | ↑ 8.83 | ↑ 2.92 | ↑ 3.95 |

**Base Model Architecture.** To assess whether the benefits of SSL4RL generalize beyond the Qwen2.5-VL series, we evaluate Gemma3-4B (Kamath et al., 2025), a recent and powerful model from Google with a distinct architecture. To ensure a fair comparison under computational constraints, we adapt our training setup by halving the batch size to 256 while keeping all other hyperparameters and training procedures unchanged.

Results presented in Table 13 provide strong evidence for the general applicability of our method. On the Gemma3 base model, our SSL4RL models yield a consistent improvement and the best average performance improvement reaches 2.88% on MMBench. The gain is particularly pronounced in Cross-instance Perception (5.76%), Logical Reasoning (4.35%), and Attribute Reasoning (4.13%). Moreover, the relative effectiveness of the individual SSL4RL tasks is remarkably consistent with our prior findings. As with Qwen2.5-VL, tasks like Rotation and Position confer the largest benefits, while the simpler Contrastive task shows more modest gains. These results suggest that SSL4RL functions as a general-purpose principle for VLMs rather than an artifact of a particular model family.

*Table 13.* Test performance (%) on MMBench. The base model is Gemma3-4B.

| Category | Model | Logical | Relation | Attribute | Coarse | Cross-Inst. | Single-Inst. | *Average* |
|---|---|---|---|---|---|---|---|---|
| Base | Gemma3-4B | 63.12 | 80.56 | 83.24 | 80.43 | 68.65 | 78.85 | 78.30 |
| | Rotation | 67.47 | 81.16 | 86.12 | 83.27 | 72.35 | 82.09 | 81.38 |
| SSL4RL | Jigsaw | 63.77 | 84.39 | 86.69 | 82.99 | 73.72 | 80.67 | 81.10 |
| | Contrastive | 63.89 | 81.94 | 87.37 | 82.30 | 71.79 | 81.70 | 80.75 |
| | Position | 64.81 | 82.26 | 86.54 | 83.08 | 74.41 | 80.89 | 81.15 |
| *Maximal Improvement* | | ↑ 4.35 | ↑ 3.83 | ↑ 4.13 | ↑ 2.84 | ↑ 5.76 | ↑ 3.24 | ↑ 3.08 |

### 4.4.3. TASK DIFFICULTY SCALING

**Varying Difficulty within Each Task.** To investigate the role of task difficulty, we intensify the corruption strategy for each SSL4RL task. Specifically, for Position and Jigsaw,

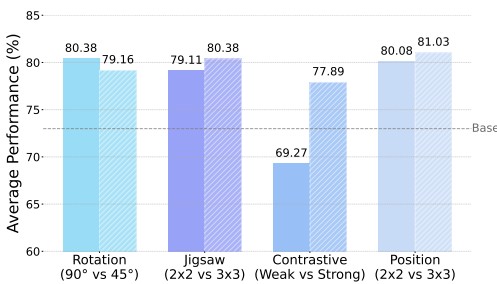

*Figure 5.* Test accuracy (%) of SSL4RL on MMBench, varying task difficulty. More results in Appendix B.

we increase the crop granularity from 2 to 3. For Contrastive, we generate harder positive samples by raising the augmentation probability from 0.2 to 0.8. For Rotation, we refine the rotation angle interval from 90° to 45°.

The impact of task difficulty varies considerably across different SSL tasks, as evidenced by Figure 5 (full results on MMBench and SEED-Bench in Appendix B). The performances of Contrastive show a marked improvement upon increasing its difficulty, elevated from 69.27% to 77.89% on MMBench. Conversely, the benefits are marginal for Position and even detrimental for Rotation and Jigsaw. A plausible explanation is that the Contrastive task's inherent simplicity as a binary discrimination problem means that raising the difficulty compels the model to learn more robust and informative features. In contrast, exacerbating the difficulty of already challenging tasks like Jigsaw might induce over-specialization to the SSL objective, resulting in a negative transfer where the learned representations are counterproductive for downstream reasoning.

**Harder SSL4RL Tasks.** Beyond adjusting the difficulty level within task, we further design two harder tasks: 1) *Mask Identification* (He et al., 2022; Chen et al., 2023; 2024c). A harder Position task that shifts from spatial prediction to identifying masked details among multiple candidates. 2) *Hard-Negative Contrastive* (Xuan et al., 2020; Ge et al., 2021). A more challenging Contrastive task that replaces random negatives with semantically similar images retrieved by DINOv2 (Oquab et al., 2023).

We evaluate the harder tasks on both 3B and 7B models. On 3B models (Table 14), for the Mask task (harder), the performance is inferior to the Position task (58.52% vs. 60.39%). We also observe significant reward collision and a lack of convergence during training, suggesting that this task may exceed the learning capacity of a 3B model. While the Hard-Negative Contrastive task (harder) outperforms the Contrastive task (59.43% vs. 54.11%), it still fails to surpass the Position task. In contrast, on 7B models (Table 15), harder tasks outperform easier ones by a substantial margin. Hard-Negative Contrastive exceeds standard Contrastive by 2.97% and Mask improves over Position by 1.3% on average. These results indicate that a SSL4RL task's effectiveness

*Table 14.* Test accuracy (%) of harder tasks for 3B models.

| Category | Model | MMBench | SEED-Bench | V* | RealWorldQA | BLINK | MME | *Average* |
|---|---|---|---|---|---|---|---|---|
| Base | Qwen2.5-VL-3B | 72.99 | 60.83 | 59.16 | 52.67 | 42.13 | 32.41 | 53.36 |
| SSL4RL-3B | Position | **80.08** | **69.77** | **68.06** | 59.86 | **46.39** | **38.19** | **60.39** |
| | **Mask** | 78.47 | 65.61 | 63.87 | **60.39** | 44.76 | 38.04 | 58.52 |
| | Contrastive | 69.27 | 61.90 | 60.20 | 58.03 | 45.13 | 30.17 | 54.11 |
| | **Hard-Contrastive** | 79.60 | 68.14 | 65.44 | 59.21 | 46.34 | 37.88 | 59.43 |

*Table 15.* Test accuracy (%) of harder tasks for 7B models.

| Category | Model | MMBench | SEED-Bench | V* | RealWorldQA | BLINK | MME | *Average* |
|---|---|---|---|---|---|---|---|---|
| Base | Qwen2.5-VL-7B | 86.37 | 74.70 | 73.29 | 65.88 | 45.50 | 45.59 | 65.22 |
| SSL4RL-7B | Position | 86.25 | 75.56 | 77.48 | 67.45 | 46.76 | 47.57 | 66.84 |
| | **Mask** | **86.85** | **76.83** | 78.01 | **68.88** | 49.28 | **49.03** | 68.14 |
| | Contrastive | 86.25 | 75.27 | 76.96 | 66.66 | 45.60 | 45.44 | 66.03 |
| | **Hard-Contrastive** | **87.78** | 75.79 | **79.05** | **70.58** | **52.23** | 48.61 | **69.00** |

is coupled with model capacity, where larger models can benefit from harder tasks while smaller models struggle with excessive complexity.

> **⚲ Takeaway**
>
> **Data Scaling Potential**. SSL4RL demonstrates promising data-scaling behavior, with performance improving monotonically as datasets grow larger and more diverse.
>
> **Robustness to Base Model Variation**. SSL4RL demonstrates consistent performance gains across different architectures and sizes of the base model.
>
> **Goldilocks Principle of Task Difficulty**. The effectiveness of an SSL task is contingent on its difficulty being appropriately matched to the model's capacity.

## 5. Conclusions

We have introduced SSL4RL, a framework that repurposes self-supervised tasks as verifiable reinforcement learning rewards for vision–language models. Our study shows that SSL4RL improves multimodal reasoning and vision-centric perception, with potentials on open-ended scenarios and stronger resilience to visual perturbations. These findings suggest a broader principle: with proper task selection, verifiable and scalable self-supervised signals can drive alignment and reasoning in VLMs without reliance on external verifiers, judges, or costly human labels. Looking forward, SSL4RL opens a path toward safer and more capable multimodal foundation models by unifying the strengths of self-supervision and reinforcement learning.

## Acknowledgements

Yisen Wang is supported by National Natural Science Foundation of China (92370129, 62376010), Beijing Major Science and Technology Project under Contract no. Z251100008425006, Beijing Nova Program (20230484344, 20240484642), and State Key Laboratory of General Artificial Intelligence.

## Impact Statement

This paper presents work whose goal is to address the tendency of vision-language models to rely on linguistic shortcuts rather than grounded visual evidence, and to propose a general framework that uses self-supervised learning objectives as verifiable reward signals for reinforcement learning. We believe its core contribution is methodological, providing a scalable alternative to human- or model-based reward design for multimodal alignment. At this stage, we do not identify specific societal impacts beyond the potential to improve the robustness and reliability of multimodal models.

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

## The Use of Large Language Models (LLMs)

In this work, LLMs are primarily employed for polishing the language of the manuscript to ensure grammatical correctness and coherence. Importantly, all conceptual development, theoretical analysis, experimental design, and result interpretation are conducted independently by the authors. The use of LLMs is strictly limited to auxiliary tasks, ensuring that the scientific contributions of this paper remain entirely unaffected by such tools.

## A. Detailed Results of MMBench and SEED-Bench

In this section, we present the detailed leaf task results of MMBench and SEED-Bench from Table 16 to Table 19.

*Table 16.* Test performance (%) of 3B models on MMBench downstream tasks. IR: Identity Reasoning, PPR: Physical Property Reasoning, FR: Function Reasoning, OL: Object Localization, SIU: Structuralized Imagetext Understanding, AtR: Attribute Recognition, FP: Future Prediction, SR: Spatial Relationship, IS: Image Scene, IQ: Image Quality, Ace: Action Recognition, AC: Attribute Comparison, IT: Image Topic, NR: Nature Relation, PR: Physical Relation, SR: Social Relation, CR: Celebrity Recognition, IS: Image Style, OCR: OCR, IE: Image Emotion.

| Category | Model | IR | PPR | FR | OL | SIU | AtR | FP | SR | IS | IQ | Ace | AC | IT | NR | PR | SR | CR | IS | OCR | IE | *Average* |
|---|---|---|---|---|---|---|---|---|---|---|---|---|---|---|---|---|---|---|---|---|---|---|
| Base | Qwen2.5-VL-3B | 86.36 | 60.27 | 83.22 | 63.17 | 71.99 | 81.44 | 51.54 | 45.20 | 91.15 | 46.67 | 84.65 | 63.12 | 82.86 | 64.25 | 47.87 | 38.37 | 89.39 | 82.55 | 94.23 | 64.50 | 72.99 |
| SSL4RL-3B | Rotation | 97.16 | 66.67 | 87.83 | 63.49 | 75.53 | 85.23 | 56.15 | 56.50 | 95.82 | 55.33 | 89.30 | 68.79 | 87.14 | 70.39 | 64.89 | 88.37 | 95.45 | 87.26 | 94.87 | 75.50 | 80.38 |
| | Jigsaw | 91.48 | 64.38 | 85.20 | 61.59 | 73.40 | 89.02 | 52.31 | 53.11 | 93.86 | 56.00 | 86.51 | 63.83 | 87.86 | 64.25 | 59.57 | 81.98 | 94.95 | 85.38 | 91.67 | 66.50 | 77.82 |
| | Contrastive | 90.34 | 52.05 | 73.03 | 60.32 | 69.15 | 78.03 | 53.08 | 32.77 | 88.45 | 42.00 | 81.40 | 60.99 | 82.14 | 36.87 | 57.45 | 81.40 | 90.40 | 61.79 | 85.26 | 52.50 | 69.27 |
| | Position | 95.45 | 64.38 | 86.84 | 66.03 | 78.37 | 86.36 | 56.92 | 42.37 | 96.31 | 58.00 | 89.77 | 67.38 | 86.43 | 68.16 | 60.64 | 85.47 | 94.95 | 91.51 | 94.23 | 78.50 | 80.08 |
| | Combination | 95.45 | 61.64 | 82.24 | 68.25 | 77.66 | 89.77 | 54.62 | 45.76 | 94.10 | 55.33 | 90.23 | 63.83 | 87.86 | 61.45 | 54.26 | 90.12 | 94.19 | 89.15 | 94.87 | 67.50 | 78.77 |

*Table 17.* Test performance (%) of 7B models on MMBench downstream tasks. IR: Identity Reasoning, PPR: Physical Property Reasoning, FR: Function Reasoning, OL: Object Localization, SIU: Structuralized Imagetext Understanding, AtR: Attribute Recognition, FP: Future Prediction, SR: Spatial Relationship, IS: Image Scene, IQ: Image Quality, Ace: Action Recognition, AC: Attribute Comparison, IT: Image Topic, NR: Nature Relation, PR: Physical Relation, SR: Social Relation, CR: Celebrity Recognition, IS: Image Style, OCR: OCR, IE: Image Emotion.

| Category | Model | IR | PPR | FR | OL | SIU | AtR | FP | SR | IS | IQ | Ace | AC | IT | NR | PR | SR | CR | IS | OCR | IE | *Average* |
|---|---|---|---|---|---|---|---|---|---|---|---|---|---|---|---|---|---|---|---|---|---|---|
| Base | Qwen2.5-VL-7B | 98.30 | 66.67 | 92.11 | 74.60 | 82.98 | 88.64 | 70.00 | 76.27 | 97.05 | 58.00 | 92.09 | 85.11 | 91.43 | 88.83 | 69.15 | 92.44 | 97.22 | 94.81 | 96.15 | 82.00 | 86.37 |
| SSL4RL-7B | Rotation | 97.73 | 71.23 | 92.43 | 71.11 | 89.72 | 90.15 | 67.69 | 78.53 | 97.79 | 61.33 | 92.56 | 94.33 | 90.71 | 85.47 | 71.28 | 93.60 | 96.97 | 96.23 | 97.44 | 83.50 | 87.50 |
| | Jigsaw | 97.73 | 74.89 | 91.45 | 75.24 | 86.52 | 93.56 | 73.85 | 74.01 | 97.54 | 60.67 | 91.63 | 87.94 | 90.00 | 89.39 | 70.21 | 91.86 | 97.47 | 94.81 | 97.44 | 85.50 | 87.73 |
| | Contrastive | 98.30 | 63.01 | 92.43 | 74.29 | 85.46 | 85.61 | 70.00 | 77.97 | 97.30 | 63.33 | 93.49 | 84.40 | 88.57 | 85.47 | 67.02 | 91.28 | 97.47 | 95.75 | 93.59 | 84.50 | 86.25 |
| | Position | 98.30 | 63.01 | 92.43 | 74.29 | 85.46 | 85.61 | 70.00 | 77.97 | 97.30 | 63.33 | 93.49 | 84.40 | 88.57 | 85.47 | 67.02 | 91.28 | 97.47 | 95.75 | 93.59 | 84.50 | 86.25 |
| | Combination | 97.73 | 73.97 | 92.11 | 71.43 | 90.43 | 95.08 | 70.77 | 74.58 | 97.79 | 61.33 | 93.49 | 81.56 | 92.14 | 89.39 | 68.09 | 93.02 | 97.47 | 93.87 | 96.79 | 84.50 | 85.78 |

*Table 18.* Test performance (%) of 3B models on SEED-Bench downstream tasks. TU: Text Understanding, VR: Visual Reasoning, SU: Scene Understanding, IId: Instance Identity, IIn: Instance Interaction, IA: Instance Attributes, IL: Instance Location, SR: Spatial Relation, IC: Instances Counting.

| Category | Model | TU | VR | SU | IId | IIn | IA | IL | SR | IC | *Average* |
|---|---|---|---|---|---|---|---|---|---|---|---|
| Base | Qwen2.5-VL-3B | 41.67 | 53.78 | 60.35 | 63.24 | 64.95 | 62.87 | 58.79 | 51.60 | 60.52 | 60.83 |
| SSL4RL | Rotation | 45.24 | 73.41 | 73.65 | 72.80 | 67.01 | 71.03 | 61.76 | 54.03 | 64.12 | 69.10 |
| | Jigsaw | 48.81 | 69.79 | 70.30 | 71.65 | 63.92 | 70.19 | 62.68 | 53.12 | 62.93 | 67.67 |
| | Contrastive | 28.57 | 67.07 | 67.10 | 68.38 | 64.95 | 61.22 | 63.70 | 51.75 | 54.03 | 61.90 |
| | Position | 52.38 | 70.69 | 73.56 | 72.75 | 62.89 | 72.51 | 64.62 | 55.25 | 64.20 | 69.77 |

*Table 19.* Test performance (%) of 7B models on SEED-Bench downstream tasks. TU: Text Understanding, VR: Visual Reasoning, SU: Scene Understanding, IId: Instance Identity, IIn: Instance Interaction, IA: Instance Attributes, IL: Instance Location, SR: Spatial Relation, IC: Instances Counting.

| Category | Model | TU | VR | SU | IId | IIn | IA | IL | SR | IC | *Average* |
|---|---|---|---|---|---|---|---|---|---|---|---|
| Base | Qwen2.5-VL-7B | 72.62 | 77.95 | 77.99 | 77.44 | 75.26 | 76.19 | 71.98 | 62.56 | 69.55 | 74.70 |
| SSL4RL-7B | Rotation | 76.19 | 78.25 | 78.59 | 77.94 | 73.20 | 76.90 | 73.11 | 64.23 | 69.76 | 75.33 |
| | Jigsaw | 70.24 | 79.15 | 78.44 | 77.66 | 76.29 | 76.58 | 73.01 | 63.77 | 69.27 | 75.05 |
| | Contrastive | 71.43 | 78.85 | 78.28 | 78.32 | 76.29 | 76.47 | 72.70 | 64.69 | 70.33 | 75.27 |
| | Position | 70.24 | 80.66 | 78.82 | 77.44 | 71.13 | 77.97 | 72.19 | 64.38 | 69.39 | 75.56 |

## B. Detailed Results of the Task Difficulty Ablation Study

From Table 20 to Table 23, we provide detailed experimental results for the ablation study on task difficulty of MMBench and SEED-Bench.

*Table 20.* Test performance (%) of 3B models trained with different task difficulties on MMBench downstream tasks. IR: Identity Reasoning, PPR: Physical Property Reasoning, FR: Function Reasoning, OL: Object Localization, SIU: Structuralized Imagetext Understanding, AtR: Attribute Recognition, FP: Future Prediction, SR: Spatial Relationship, IS: Image Scene, IQ: Image Quality, Ace: Action Recognition, AC: Attribute Comparison, IT: Image Topic, NR: Nature Relation, PR: Physical Relation, SR: Social Relation, CR: Celebrity Recognition, IS: Image Style, OCR: OCR, IE: Image Emotion.

| Model | Difficulty | IR | PPR | FR | OL | SIU | AtR | FP | SR | IS | IQ | Ace | AC | IT | NR | PR | SR | CR | IS | OCR | IE | *Average* |
|---|---|---|---|---|---|---|---|---|---|---|---|---|---|---|---|---|---|---|---|---|---|---|
| Qwen2.5-VL-3B | – | 86.36 | 60.27 | 83.22 | 63.17 | 71.99 | 81.44 | 51.54 | 45.20 | 91.15 | 46.67 | 84.65 | 63.12 | 82.86 | 64.25 | 47.87 | 38.37 | 89.39 | 82.55 | 94.23 | 64.50 | 72.99 |
| Rotation | 90-degree | **97.16** | **66.67** | **87.83** | 63.49 | 75.53 | 85.23 | 56.15 | **56.50** | 95.82 | 55.33 | 89.30 | 68.79 | 87.14 | 70.39 | 64.89 | 88.37 | 95.45 | 87.26 | **94.87** | 75.50 | 80.38 |
| | 45-degree | 93.75 | 64.84 | **87.83** | 66.67 | 74.11 | 90.15 | 58.46 | 46.33 | 95.33 | 53.33 | 88.84 | 65.25 | 87.14 | 65.92 | 54.26 | 90.12 | 93.94 | 86.79 | 92.31 | 70.50 | 79.16 |
| Jigsaw | 2x2 | 91.48 | 64.38 | 85.20 | 61.59 | 73.40 | 89.02 | 52.31 | 53.11 | 93.86 | 56.00 | 86.51 | 63.83 | 87.86 | 64.25 | 59.57 | 81.98 | 94.95 | 85.38 | 91.67 | 66.50 | 77.82 |
| | 3x3 | 93.75 | 60.27 | 82.57 | 60.95 | 65.60 | 84.47 | 49.23 | 51.41 | 91.89 | 56.67 | 83.26 | 65.25 | 82.86 | 66.48 | 50.00 | 78.49 | 85.35 | 85.85 | 88.46 | 72.00 | 75.12 |
| Contrastive | Weak | 90.34 | 52.05 | 73.03 | 60.32 | 69.15 | 78.03 | 53.08 | 32.77 | 88.45 | 42.00 | 81.40 | 60.99 | 82.14 | 36.87 | 57.45 | 81.40 | 90.40 | 61.79 | 85.26 | 52.50 | 69.27 |
| | Strong | 94.89 | 64.84 | 82.24 | 63.49 | 77.30 | 86.36 | 51.54 | 44.07 | 95.82 | 47.33 | 89.30 | 70.21 | 85.71 | 59.78 | 52.13 | 86.63 | 94.44 | 87.74 | 92.31 | 70.50 | 77.89 |
| Position | 2x2 | 95.45 | 64.38 | 86.84 | 66.03 | 78.37 | 86.36 | 56.92 | 42.37 | 96.31 | 58.00 | 89.77 | 67.38 | 86.43 | 68.16 | 60.64 | 85.47 | 94.95 | 91.51 | 94.23 | 78.50 | 80.08 |
| | 3x3 | **97.16** | 63.47 | 87.50 | 66.35 | 75.53 | 88.64 | **60.00** | 52.54 | 95.33 | 58.67 | 86.51 | **76.60** | 88.57 | 70.39 | 58.51 | 88.95 | 95.96 | 92.45 | 93.59 | 77.50 | **81.03** |

*Table 21.* Test performance (%) of 7B models trained with different task difficulties on MMBench downstream tasks. IR: Identity Reasoning, PPR: Physical Property Reasoning, FR: Function Reasoning, OL: Object Localization, SIU: Structuralized Imagetext Understanding, AtR: Attribute Recognition, FP: Future Prediction, SpR: Spatial Relationship, ISc: Image Scene, IQ: Image Quality, Ace: Action Recognition, AC: Attribute Comparison, IT: Image Topic, NR: Nature Relation, PR: Physical Relation, SoR: Social Relation, CR: Celebrity Recognition, ISt: Image Style, OCR: OCR, IE: Image Emotion.

| Model | Difficulty | IR | PPR | FR | OL | SIU | AtR | FP | SpR | ISc | IQ | Ace | AC | IT | NR | PR | SoR | CR | ISt | OCR | IE | *Average* |
|---|---|---|---|---|---|---|---|---|---|---|---|---|---|---|---|---|---|---|---|---|---|---|
| Base | Qwen2.5-VL-7B | **98.30** | 66.67 | 92.11 | 74.60 | 82.98 | 88.64 | 70.00 | 76.27 | 97.05 | 58.00 | 92.09 | 85.11 | **91.43** | 88.83 | 69.15 | 92.44 | 97.22 | 94.81 | 96.15 | 82.00 | **86.37** |
| Jigsaw | 3x3 | **98.30** | 69.41 | 92.11 | 76.83 | 86.88 | 90.91 | 70.77 | 79.10 | 97.54 | 62.00 | 92.56 | 87.94 | 90.00 | 89.94 | 70.21 | 93.02 | 97.47 | 95.28 | 98.72 | 84.50 | 86.17 |
| | 4x4 | 97.73 | 66.21 | 92.43 | 75.24 | 86.17 | 92.80 | 73.08 | 76.27 | 97.79 | 62.67 | 92.09 | 87.94 | 90.71 | 87.71 | 69.15 | 91.86 | 97.47 | 94.81 | 97.44 | 85.00 | 85.73 |
| | 5x5 | 97.73 | 67.58 | 92.76 | 74.29 | 85.46 | 90.15 | 73.08 | 79.10 | 97.05 | 61.33 | 93.02 | 87.94 | 90.71 | 89.94 | 70.21 | 91.28 | 97.98 | 95.75 | 95.51 | 84.00 | 85.74 |
| Position | 3x3 | 95.45 | 68.49 | 93.09 | 77.46 | 88.30 | 94.70 | 73.08 | 77.40 | 97.79 | 62.67 | 94.42 | 89.36 | 90.71 | 87.15 | 67.02 | 89.53 | 97.47 | 94.34 | 97.44 | 81.50 | 85.87 |
| | 4x4 | 97.73 | 64.84 | **94.74** | 74.92 | 88.65 | 95.45 | 73.08 | 72.32 | 97.79 | 58.00 | 94.42 | 82.98 | 89.29 | 88.83 | 69.15 | 90.12 | 98.23 | 94.34 | 98.08 | 82.50 | 85.27 |
| | 5x5 | 97.73 | **72.15** | 93.75 | 75.24 | **90.07** | 95.83 | 70.00 | 67.80 | 98.53 | 63.33 | 95.81 | 90.07 | 90.71 | 91.62 | 67.02 | 91.86 | 98.23 | 95.28 | 95.51 | 81.00 | 86.08 |

*Table 22.* Test performance (%) of 3B models trained with different task difficulties on SEED-Bench downstream tasks. IC: Instances Counting, IA: Instance Attributes, SU: Scene Understanding, IId: Instance Identity, IIn: Instance Interaction, VR: Visual Reasoning, IL: Instance Location, SR: Spatial Relation, TU: Text Understanding.

| Model | Difficulty | IC | IA | SU | IId | IIn | VR | IL | SR | TU | *Average* |
|---|---|---|---|---|---|---|---|---|---|---|---|
| Qwen2.5-VL-3B | – | 60.52 | 62.87 | 60.35 | 63.24 | 64.95 | 53.78 | 58.79 | 51.60 | 41.67 | 60.83 |
| Rotation | 90-degree | 64.12 | 71.03 | 73.65 | 72.80 | 67.01 | 73.41 | 61.76 | 54.03 | 45.24 | 69.10 |
| | 45-degree | 60.60 | 69.26 | 72.70 | 72.53 | 68.04 | 72.81 | 63.60 | 55.25 | 38.10 | 67.81 |
| Jigsaw | 2x2 | 62.93 | 70.19 | 70.30 | 71.65 | 63.92 | 69.79 | 62.68 | 53.12 | 48.81 | 67.67 |
| | 3x3 | 61.22 | 67.58 | 69.28 | 68.87 | **69.07** | 64.35 | 61.35 | 51.14 | 48.81 | 65.66 |
| Contrastive | Weak | 54.03 | 61.22 | 67.10 | 68.38 | 64.95 | 67.07 | 63.70 | 51.75 | 28.57 | 61.90 |
| | Strong | 57.54 | 64.75 | 70.68 | 70.56 | 67.01 | 70.39 | 63.39 | 55.86 | 29.76 | 65.00 |
| Position | 2x2 | 64.20 | 72.51 | 73.56 | 72.75 | 62.89 | 70.69 | 64.62 | 55.25 | **52.38** | 69.77 |
| | 3x3 | 61.87 | **72.53** | **73.84** | **73.89** | **69.07** | 74.02 | 64.62 | 58.30 | 44.05 | **69.80** |

## C. Robustness Analysis

### C.1. Evaluation on Perturbed MMBench

To evaluate our model's robustness, we conduct a comprehensive evaluation under various image perturbations. We design two levels of perturbation—weak and strong—to assess the models' resilience to visual corruptions. The weak perturbation applies a series of common image transformations, each with a probability of 0.5, including color jittering, random conversion to grayscale, Gaussian blur, and horizontal flipping. The strong perturbation builds upon the weak one by incorporating an additional multi-crop strategy. This strategy generates two 224×224 pixel crops per image via randomly

*Table 23.* Test performance (%) of 7B models trained with different task difficulties on SEED-Bench downstream tasks. IC: Instances Counting, IA: Instance Attributes, SU: Scene Understanding, IId: Instance Identity, IIn: Instance Interaction, VR: Visual Reasoning, IL: Instance Location, SR: Spatial Relation, TU: Text Understanding.

| Category | Model | TU | VR | SU | IId | IIn | IA | IL | SR | IC | *Average* |
|---|---|---|---|---|---|---|---|---|---|---|---|
| Base | Qwen2.5-VL-7B | **72.62** | **77.95** | 77.99 | 77.44 | 75.26 | 76.19 | 71.98 | 62.56 | 69.55 | **74.70** |
| Jigsaw | 3x3 | 69.88 | 77.01 | **78.78** | 77.88 | **76.29** | **79.46** | 71.57 | **64.69** | 73.81 | 74.37 |
| | 4x4 | 69.72 | 76.98 | 77.83 | 78.15 | 71.13 | 78.85 | 72.29 | 62.10 | 72.62 | 73.30 |
| | 5x5 | 69.84 | 76.43 | 78.34 | 78.15 | 71.13 | 78.85 | **73.21** | 63.17 | 70.24 | 73.26 |
| Position | 3x3 | 68.82 | 77.93 | 78.53 | **78.75** | 74.23 | 78.55 | 73.01 | 64.08 | 76.19 | 74.45 |
| | 4x4 | 68.98 | 77.41 | 77.58 | 77.33 | **76.29** | 78.55 | 71.47 | 61.64 | 76.19 | 73.94 |
| | 5x5 | 69.27 | 77.39 | 78.59 | 77.72 | 73.20 | 79.15 | **73.21** | 63.47 | **77.38** | 74.38 |

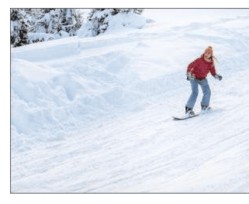
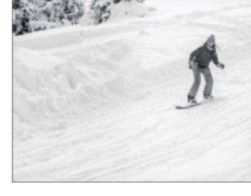
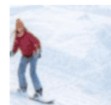

Original Iamge              Weak perturbation              Strong perturbation

Question: Identify the question that Madelyn and Tucker's experiment can best answer.

A. Does Madelyn's snowboard slide down a hill in less time when it has a thin layer of wax or a thick layer of wax?

B. Does Madelyn's snowboard slide down a hill in less time when it has a layer of wax or when it does not have a layer of wax?

*Figure 6.* Perturbation examples on MMBench.

resized cropping (scale range [0.08, 1.0]), presenting a more significant challenge to the model's perception. Examples of these perturbed images are provided in Figure 6.

We evaluate our SSL4RL models and the base model on the perturbed MMBench benchmarks. As shown in Table 8 and Table 9, **our method consistently demonstrates superior robustness across both perturbation levels**. For instance, under weak perturbation, our SSL4RL-Rotation model achieves an average score of 72.11%, outperforming the base model at 66.80%. This performance gap persists under strong perturbation, where our SSL4RL-Position model scores 64.37% compared to the base model's 59.62%. These results confirm that our SSL4RL strategy effectively enhances the model's robustness to photometric variations.

### C.2. Evaluation on VLM Robustness Benchmarks

To further investigate the resilience of SSL4RL to visual perturbations, we conduct an evaluation on a comprehensive robustness benchmark for VLMs Ishmam et al. (2025). The benchmark encompasses 213,000 synthetically corrupted images—derived from 3,000 originals—alongside 16,000 associated question-answer pairs. A suite of distinct corruption functions (e.g., motion blur, snow, brightness transforms) is applied at five discrete severity levels, with the intensity of distortion dictated by function-specific parameters. As shown in Table 24, our SSL4RL models outperform the base model across nearly all corruption functions and severity levels. Notably, the performance advantage becomes more pronounced at higher corruption levels. For instance, our models surpass the base model by 5.37% on Contrast-Level5 (34.01% vs. 28.64%) and by 3.46% on Elastic-Level5 (37.16% vs. 33.70%).

## D. Comparison with Verifier-Driven RL using Downstream Golden Rewards

To contextualize the performance of our SSL4RL method, we include a strong, task-specific baseline *VLM-R1* (Shen et al., 2025). It extends R1-style reinforcement learning to VLMs with rule-based reward formulation. Specifically, we randomly split downstream datasets into 60% for training and 40% for held-out testing. Then we fine-tune Qwen2.5-VL-3B using the GRPO algorithm, where the reward is computed by comparing it directly to the golden answer. We denote the tuned model

*Table 24.* Performance (%) on the VLM robustness benchmark Ishmam et al. (2025). We show detailed results on corruption functions (Brightness, Contrastive, Defocus-Blur, Elastic, JPEG-Compression, Pixelate, Saturation, Snow, Spatter, Zoom-Blur) with varying severity levels from 1 to 5.

| Level | L1 | L2 | L3 | L4 | L5 | L1 | L2 | L3 | L4 | L5 |
|---|---|---|---|---|---|---|---|---|---|---|
| | Brightness | | | | | Contrastive | | | | |
| Qwen2.5-VL-3B | 68.47 | 66.79 | 65.46 | 63.55 | 61.81 | 67.97 | 66.30 | 62.12 | 48.87 | 28.64 |
| SSL4RL-Position | **68.54** | **67.24** | **65.47** | **63.74** | **61.86** | **68.12** | **66.70** | **63.34** | **52.51** | **34.01** |
| | Defocus-Blur | | | | | Elastic | | | | |
| Qwen2.5-VL | 60.25 | 57.51 | 48.11 | 40.28 | 32.95 | 65.16 | 59.60 | 61.30 | 55.72 | 33.70 |
| SSL4RL-Position | **61.35** | **59.06** | **51.22** | **44.16** | **37.25** | **65.74** | **61.13** | **62.57** | **57.21** | **37.16** |
| | JPEG-Compression | | | | | Pixelate | | | | |
| Qwen2.5-VL-3B | 50.19 | 50.28 | 49.15 | 43.22 | 32.25 | 60.31 | 60.73 | 54.54 | 44.30 | 6.28 |
| SSL4RL-Position | **52.98** | **52.76** | **51.83** | **46.71** | **36.38** | **60.82** | **60.87** | **54.94** | **45.13** | **6.63** |
| | Saturation | | | | | Snow | | | | |
| Qwen2.5-VL-3B | **66.70** | **62.95** | **67.97** | 63.93 | 58.87 | 61.07 | 49.64 | 53.11 | 47.06 | 38.80 |
| SSL4RL-Position | 66.54 | 62.87 | 67.93 | **63.98** | **59.77** | **61.28** | **51.52** | **53.96** | **48.38** | **41.29** |
| | Spatter | | | | | Zoom-Blur | | | | |
| Qwen2.5-VL-3B | 68.42 | 61.58 | 60.26 | 56.67 | 50.37 | 56.95 | 52.06 | 49.43 | 45.72 | 42.85 |
| SSL4RL-Position | **68.48** | **62.68** | **61.20** | **58.35** | **52.68** | **57.58** | **53.55** | **50.73** | **47.33** | **44.79** |

as **Golden-3B**. Notably, the model is trained with **oracle reward signals**, and the setup simulates an ideal scenario where the reward model is perfectly aligned with the evaluation metric. The results are presented in Table 2 and Table 3, where we draw two key conclusions:

- **The Oracle Baseline has a high but finite ceiling.** Even with direct supervision from golden answers, the Golden-3B model achieves 84.93% on MMBench and 73.21% on SEED-Bench. This indicates inherent challenges in these benchmarks that are not fully solved even with ideal, verifiable rewards.

- **Our SSL4RL method is highly competitive.** The performance gap between our best SSL4RL variants and the Golden-3B oracle is relatively small (*e.g.*, 81.35% vs. 84.93% on MMBench, and 69.80% vs. 73.21% on SEED-Bench), compared to our improvements over the base model. This demonstrates that our self-supervised objectives, which require *no labeled downstream data*, can effectively close the gap to the performance driven by idealized, task-specific reward signals.

This comparison paints an encouraging picture for SSL4RL. Our method, which deliberately avoids using any downstream labels, can close much of the gap to a model trained with direct oracle signals. It suggests that the reinforcement learning process can be steered effectively by self-supervised objectives. While perfect verifiable rewards remain a powerful tool, our work shows that highly competitive performance can be achieved through a more scalable and generalizable pathway.

## E. Evaluation on Vision-Language Benchmarks

In this section, we evaluate the SSL4RL strategy on broader vision-language benchmarks, including Vision Questioning Answer (VQA) tasks and the open-ended image-caption task.

### E.1. Visual Question Answering (VQA) Tasks

Besides MMBench and SEED-Bench, we further consider four representative VQA benchmarks, V*(Wu & Xie, 2024), RealWorldQA (xAI, 2024), BLINK (Fu et al., 2024), MME-RealWorld-Lite (Zhang et al., 2024b), featuring challenging current VLMs' ability of visual perception, spatial understanding, detail capturing, real-world applications, and so on. The detailed descriptions of benchmarks are as follows:

- **MMBench** (Liu et al., 2024): A diverse benchmark with over 3,000 multiple-choice questions spanning 20 distinct

ability dimensions. All results on MMBench are reported for the DEV-EN split, showing the average performance per category.

- **SEED-Bench** (Li et al., 2023): A comprehensive benchmark with human annotations for image and video modalities. For evaluation, we select the 9 core dimensions pertaining to image understanding, which comprise 14,232 examples in total.

- **V\*** (Wu & Xie, 2024): A benchmark based on 191 high-resolution images with an average image resolution of 2246×1582. It is specifically designed to quantitatively evaluate VLMs' ability in challenging scenarios where the image contains abundant and complex information, and the visual information needed might not be easily found.

- **RealWorldQA** (xAI, 2024): A benchmark designed to evaluate the real-world visual understanding capabilities of VLMs. It assesses how well these models comprehend physical environments. The benchmark consists of 700+ images drawn from real-world scenarios, including those captured from vehicles.

- **BLINK** (Fu et al., 2024): A benchmark for VLMs that focuses on core visual perception abilities, including relative depth estimation, visual correspondence, forensics detection, and multi-view reasoning, etc. BLINK contains visual commonsense problems that humans can answer within seconds, rarely requiring domain knowledge.

- **MME-RealWorld-Lite** (Zhang et al., 2024b): A benchmark contains 13K high-quality images annotated humans, resulting in 29K question-answer pairs that cover 43 subtasks across 5 real-world scenarios, featuring the highest resolution and a targeted focus on real-world applications. For inference acceleration, we utilize its public lite version MME-RealWorld-Lite with 50 samples per task.

We consider the four SSL4RL strategies: Rotation, Jigsaw, Contrastive, and Position, under the same training and evaluation settings in Section 4. The results consistently demonstrate the effectiveness of SSL4RL. As shown in Table 1), **our method provides a substantial average performance gain of 7.27% over the base model**. The improvements are particularly pronounced on V\* (+8.90%) and RealWorldQA (+9.55%). These results underscore that SSL4RL significantly enhances the model's ability to process fine-grained details in complex, real-world images. These consistent results across a diverse set of benchmarks verify the effectiveness of the proposed SSL4RL framework.

### E.2. Open-ended Vision-Language Task: Image Captioning

Relying solely on the data itself, SSL4RL requires no human labels, external verifiers, or heuristic judges, yet produces dense and scalable reinforcement signals. Intuitively, it has potential for open-ended tasks where ground truth is ill-defined and human annotations are expensive. To evaluate SSL4RL's potential on open-ended tasks, we leverage a recent image captioning platform, CapArena (Cheng et al., 2025), which contains over 6,000 human-annotated pairwise preference battles. In CapArena, captions from a test model are compared against those from strong baseline models (GPT-4o, CogVLM-19B, or MiniCPM-8B) using an LLM (GPT-4o) as a judge, with human references provided for context. The winner is assigned +1, the loser with -1, and 0 for a draw in each pairwise comparison. This challenging task mirrors the real-world challenge of improving a model without a single clear and correct answer.

We evaluate our four SSL4RL strategies on CapArena against the base Qwen2.5-VL-3B model. For easier comparison, we present the final scores after min-max normalization in Table 6. **SSL4RL consistently improves over the base model, with the largest performance gain of 8.14 points (56.45 vs. 48.31)**. This clearly shows that the self-supervised rewards provide a meaningful learning signal even in the absence of a verifiable ground truth.

**Qualitative Insights.** We conduct a qualitative comparison in Table 25, shed light on *how* SSL4RL enhances open-ended generation:

- **Enhanced Detail Capture.** SSL4RL models correctly capture more details in the image. For instance, our SSL4RL model correctly identifies a "scoreboard" displaying "1 0 0" and a "disabled persons sign," which the base model either misinterprets or omits.

- **Improved Spatial Reasoning.** SSL4RL models tend to use more precise spatial descriptors like "*behind the fence*," "*to the left of*," and "*the center of the court*", aligning well with the spatial reasoning ability required by SSL pretext tasks.

*Table 25.* Examples: Image-Captioning Comparison on CapArena.

| Image | Caption |
|---|---|
| 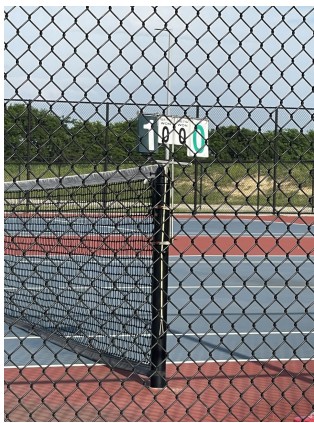 | **Qwen2.5-VL-3B:** A view of a tennis court through a chain-link fence. The court features a red and blue surface with white boundary lines. The fence has a sign indicating "100" and "WON" on the top right side. The background includes a fence and greenery, suggesting an outdoor setting. 

 **SSL4RL-Rotation:** The image depicts a tennis court viewed through a black chain-link fence. The court features a red and blue surface with white lines marking the boundaries and service boxes. A net is stretched across the center of the court, supported by black posts. Behind the fence, a scoreboard displays the score "1 0 0" indicating that one player has won all three sets. The background includes a chain-link fence enclosing the tennis court and some greenery beyond it. The sky appears clear, suggesting a sunny day. |
| 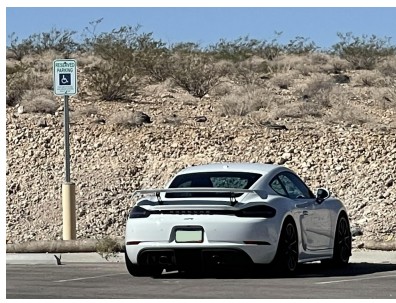 | **Qwen2.5-VL-3B:** A white Porsche sports car is parked on the side of a road in a desert-like environment. The car is positioned near a "Reserved Parking" sign, which is mounted on a pole and partially obscured by a large fallen tree trunk. The surrounding area is rocky and barren, with sparse vegetation in the background. The sky is clear and blue, indicating a sunny day. The car's sleek design and modern features are prominently displayed, emphasizing its status as a high-performance vehicle. 

 **SSL4RL-Rotation:** A white Porsche sports car is parked in a reserved parking spot on a rocky, barren landscape with sparse vegetation and a clear blue sky above. The car's sleek design and modern features are prominently displayed, contrasting with the rugged surroundings. A "Reserved Parking" sign is visible to the left of the car, indicating designated parking for individuals with disabilities. The overall scene suggests a location that could be part of a natural reserve or an eco-friendly area. |

In summary, this experiment demonstrates that SSL4RL is not limited to tasks with easily verifiable rewards. By providing a dense, automated learning signal derived from the data's intrinsic structure, our method successfully improves model performance on the complex, open-ended task of image captioning.

## F. RL Training Curves

### F.1. RL Reward Curves

Figure 7 illustrates the evolving reward signals of different SSL tasks during reinforcement learning training. Despite being intuitive for humans, these SSL tasks pose significant challenges for the base VLM. For instance, Qwen2.5-VL-3B initially achieves only about 20% accuracy on the Rotation task and nearly zero on the Jigsaw task, highlighting its limited low-level perceptual capability regarding spatial and structural image properties. Position and Rotation tasks exhibit gradual learning curves, plateauing at a reward of about 0.8 after approximately 200 epochs. For the harder Jigsaw task, the reward surges and plateaus within 20 epochs for 3B models, while 7B models present smoother reward curves, indicating a stronger learning capability of large-scale models. For the Contrastive task, models rapidly surge to 1.0 within 50 epochs due to the task's easy complexity. Figure 8 presents the reward curves of tasks with different difficulties. For the same task, increasing the difficulty slows down the learning process and lowers the final converging rewards, showing the sensitivity of the models to tasks' difficulty.

### F.2. RL Entropy Curves

Figure 9 presents the entropy trajectory of SSL4RL models during reinforcement training. In general, the entropy shows a decrease-then-increase trend. This is an emergent signature of a successful optimization process balancing two competing objectives: *specialization* and *generalization*. At the first stage, the learning rewards incentivize a lower-entropy, specialized

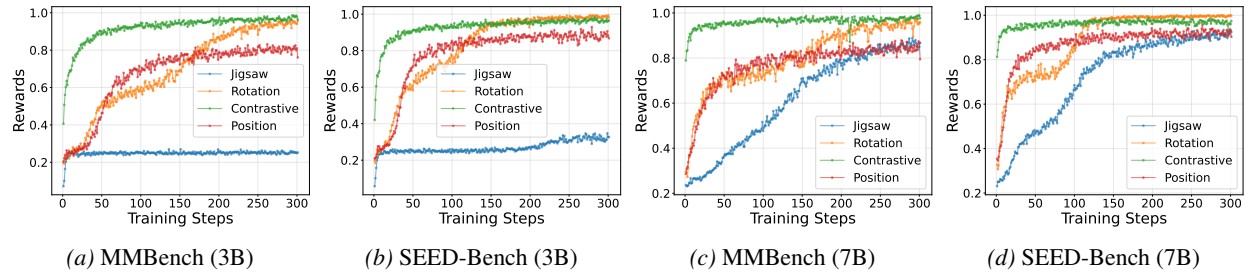

*Figure 7.* Reward curves of SSR4RL models during reinforcement learning.

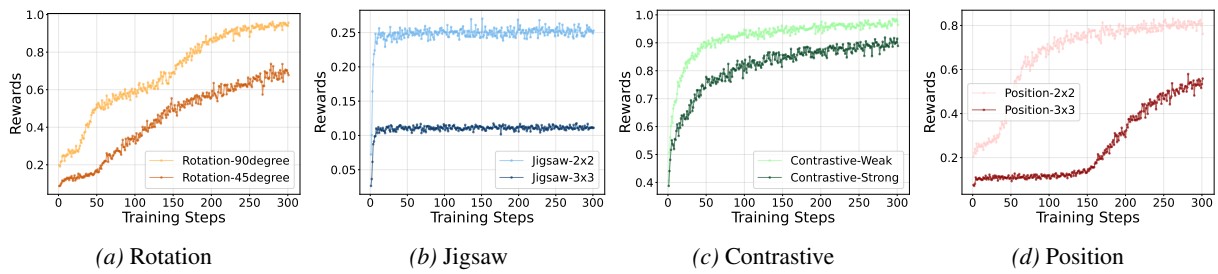

*Figure 8.* Rewards of SSL4RL 3B-models on MMBench, comparing different difficulties.

policy for high performance on the target task, resulting in a sharpening of the probability distribution. For a given prompt, the model becomes more confident in a subset of the vocabulary, directly leading to a reduction in entropy. At the second stage, to reduce the costly KL penalty and prevent mode collapse, the policy model slightly broadens probability distributions. This incentivizes a higher-entropy policy to maintain linguistic diversity and prevent catastrophic forgetting of the pre-training distribution. The two-stage entropy curves reflect a sustainable compromise between maximizing reward and preserving the foundational knowledge and generative diversity, supporting our experimental findings that SSL4RL models can generalize well to downstream vision-language tasks.

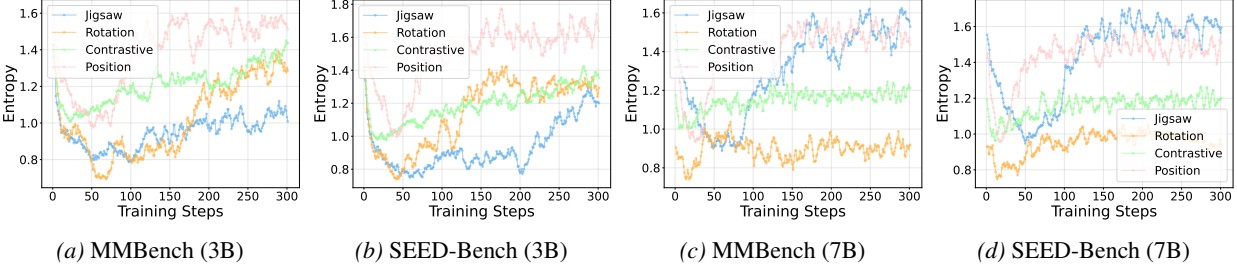

*Figure 9.* Entropy of SSR4RL models during reinforcement learning.

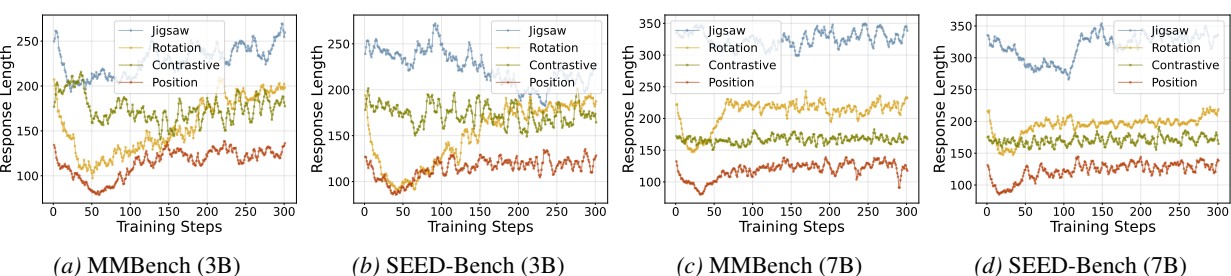

*Figure 10.* Response Lengths of SSR4RL models during reinforcement learning.

### F.3. Response Length Curves

As illustrated in Figure 10, reinforcement learning induced no substantial change in model response length, with variations remaining within a 50-token margin. This is different from DeepSeek-R1 (Guo et al., 2025), which reports longer responses featuring complex chains of thought after RL on complex mathematical and coding problems. The disparity is attributable to the nature of our SSL4RL tasks, which are perception-centric and necessitate less extensive reasoning. The second observation is that the harder Jigsaw task requires longer responses than other tasks, a correlation between task difficulty and response length that aligns with findings from Wang et al. (2025).

## G. Case Analysis and Visualization

In Figure 11, we show that SSL4RL mitigates over-reliance on linguistic priors, fostering greater dependence on visual evidence. For example, when asked about a chandelier's color, the base model defaults to a common-sense response (*e.g.*, a typical decorative color), while our model first localizes the object and then answers based on the actual appearance.

Besides, we visualize the attention maps of the baseline model, *i.e.*, Qwen2.5-VL-3B and our models on several examples from the SEED-Bench dataset (Li et al., 2023). We pick a dominant token from the questions of each example, calculate the attention map of the first generated token to that input token, and average the attention matrices of all heads and all layers of the language model. The results in Figure 12 illustrate that our models consistently display more focused attention towards the regions in the images corresponding to the selected token, which indirectly proves the superior performance of our models.

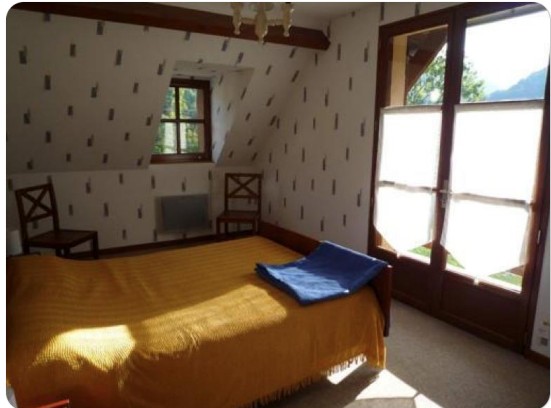

**Question:** What color is the chandelier hanging from the ceiling?
A: black     B: brown     C: white     D: gold

**Response:** Given that chandeliers are typically made of materials like glass, metal, or crystal, and considering the options provided, the most likely color for a chandelier would be gold. **The chandelier is likely to be gold because it is a common color for decorative lighting fixtures, and gold is often used in chandeliers.**

**Base Model**

**Response:** Step 1: Identify the type of lighting fixture in the image. **The image shows a chandelier hanging from the ceiling.** Step 2: Determine the color of the chandelier. **The chandelier is described as being white.** Step 3: Match the color of the chandelier with the given options. The correct option is C. white.

**Our Model**

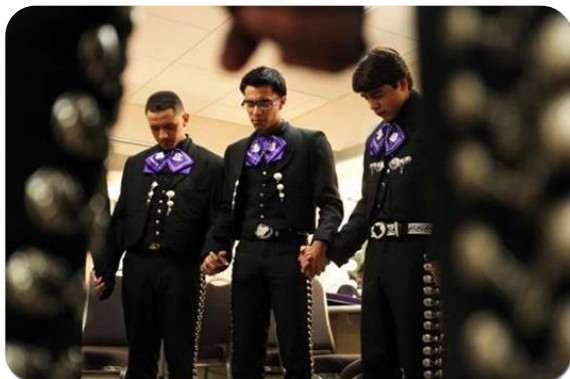

**Question:** What object is being held by one of the men in Mexican outfits, and what is he doing with it?
A: Holding a chain and swinging it
B: Holding a flower and smelling it
C: Holding a flag and waving it
D: Holding hands with another man

**Response:** B. Holding a flower and smelling it - This option implies that the man is holding a flower and smelling it, which is a common gesture in many cultures... D. Holding hands with another man - This option implies that the man is holding hands with another man, which is a common gesture in many cultures. Considering the options, **the most appropriate action for a man in a Mexican outfit would be to hold a flower and smell it,** as this is a common gesture in many cultures. <answer>B</answer>

**Base Model**

**Response:** Step 1: Analyze the image description. **The image shows three men in Mexican outfits, standing in a line and holding hands.** Step 2: Identify the specific object being held... Step 3: Determine the action being performed... Step 4: Match the action with the given options... Option D: Holding hands with another man... <answer>D</answer>

**Our Model**

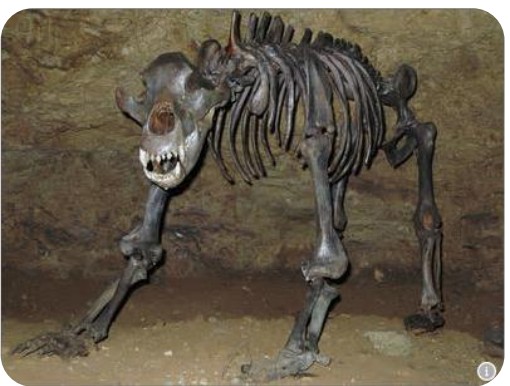

**Question:** Which trait did Ursus spelaeus have? Select the trait you can observe on the fossil.
A: rounded ears
B: brown fur covering most of its body
C: long legs

**Response:** Ursus spelaeus, also known as the cave bear, is a species of bear that lived during the Pleistocene epoch. Based on the fossil image provided, we can observe the following traits: 1. Rounded ears: **The fossil shows rounded ears, which is a characteristic of Ursus spelaeus.** 2. Brown fur covering most of its body: ... the overall appearance suggests it had brown fur. 3. Long legs: The fossil shows long legs... Given these observations, **the trait that can be clearly observed on the fossil is the rounded ears.**

**Base Model**

**Response:** To determine which trait Ursus spelaeus had based on the fossil, let's analyze the given options: A. Rounded ears - This trait is **not directly observable on the fossil provided. The fossil shows a skull and skeleton, but not the ears.** B. Brown fur covering most of its body - This trait is also not directly observable on the fossil... C. Long legs - This trait can be observed on the fossil.... The correct trait that Ursus spelaeus had, as observable on the fossil, is: C, long legs.

**Our Model**

*Figure 11.* Instances of VLMs' loss on image information. After receiving textual instructions, VLMs may be more inclined to rely on the encoded textual knowledge for reasoning rather than carefully observing the content of the image.

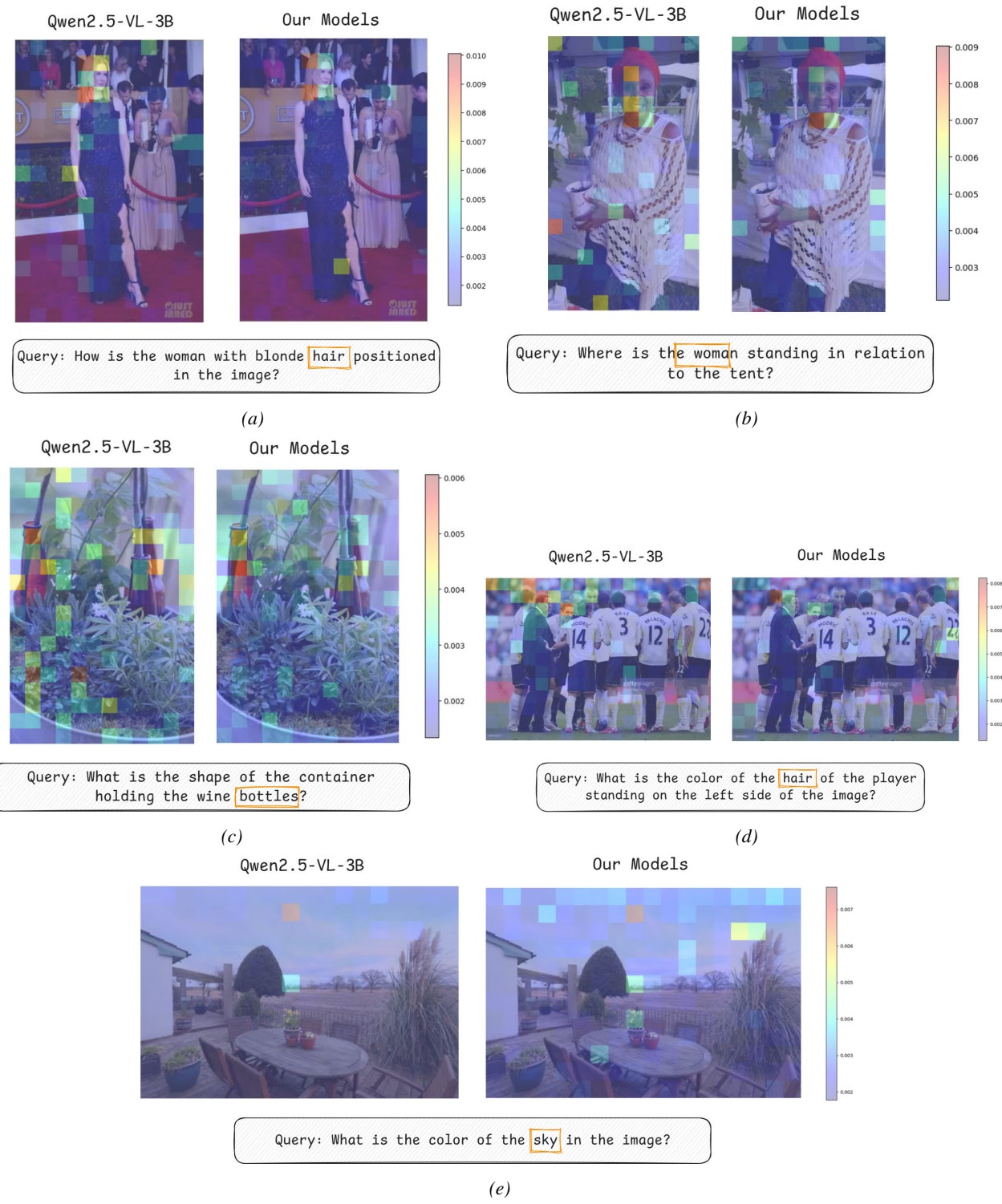

*Figure 12.* Comparisons of Attention Maps.

# H. SSL Task Examples

In this section, we show a specific instance of Rotation, Jigsaw, Contrastive, and Position tasks to illustrate the SSL task design.

---

**Rotation Example**

**Query**: These are two images. The second image is a rotated version of the first image. Please determine how many degrees the second image has been rotated **counter-clockwise** relative to the first image.
You must reason step-by-step and then provide the final answer. The output **must strictly follow** this format: <think>your reasoning here </think><answer>number_of_degrees</answer>.

**Answer**: 270

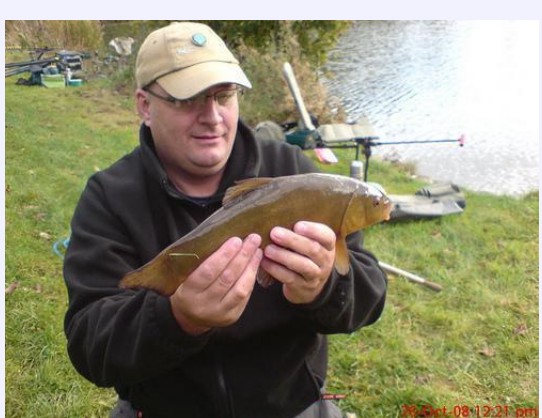
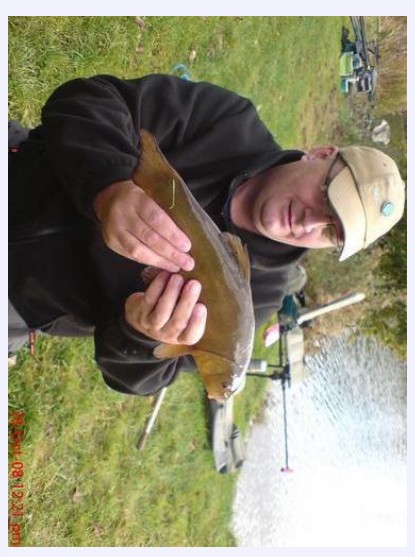

---

**Jigsaw Example**

**Query**: <image><image><image><image>
<image><image><image><image><image>
The provided images represent 9 parts of an original image, divided into a 3x3 grid.
Your task is to determine the correct order of these parts to reconstruct the original image. Starting from the top-left corner, proceed row by row, from left to right and top to bottom, to arrange the parts.
The output should be a string of numbers, separated by a comma, where each number corresponds to the original position of the patches in the restored image. For instance, "3,1,9,2,8,5,4,6,7" would indicate the positions of the patches in the correct order.
Before providing the final result, you must reason through the puzzle step by step. Consider the relative placement of each part and how they fit together.
Your answer should strictly follow this format:
<think>your step-by-step reasoning here</think><answer>order</answer>

**Answer**: 2,7,6,1,3,5,9,8,4

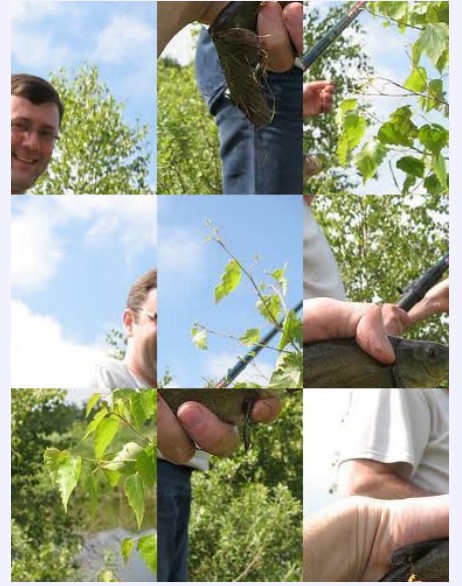

---

**Contrastive Example**

**Query**: <image><image>
The provided images are augmentations of the same original image or two different images. The augmentations may include random cropping, color adjustments, grayscale conversion, blurring, and flipping. Please think step-by-step and determine if these two images are possibly derived from the same original image. If the provided images are from the same original image, respond with "positive"; if they correspond to different original images, respond with "negative".
Your answer should strictly follow this format:
<think>your step-by-step reasoning here</think><answer>positive/negative</answer>

**Answer**: positive

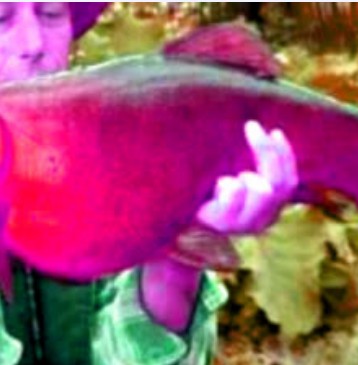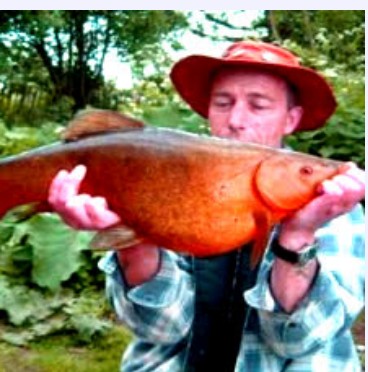

---

**Position Example**

**Query**: <image><image>
The second image in an augmented version of a crop in the first image. The augmentations may include grayscale, color jitter, solarization, etc. Please determine which part of the first image the second image is from. The second image is partitioned into 3x3 parts, and the first image can be only from one of the parts, but cannot be across two parts. The answer should be in the format of x/y, where x is the row number (from top to bottom) and y is the column number (from left to right). For example, 1/1 indicates the top-left part, and 1/3 indicates the top-right part. Both x and y may take values from 1 to 3.
Your answer should strictly follow this format:
<think>your step-by-step reasoning here</think><answer>x/y</answer>

**Answer**: 3/3

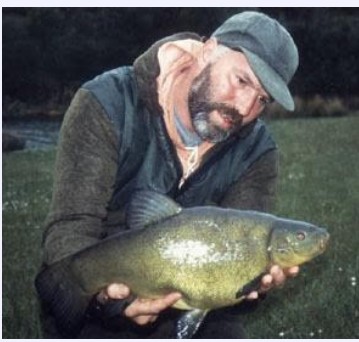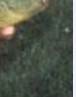

---

## I. Downstream Benchmark Example

In this section, we show a specific instance of Rotation, Jigsaw, Contrastive, and Position tasks to illustrate the SSL task design.

**Imagenet-Completion Example**

**Query**: <image>This is an image containing an object. Please identify the species of the object based on the image. The output answer format should be as follows: <think>... </think><answer>species name</answer>. Please strictly follow the format.
**Answer**: tench, Tinca, tinca

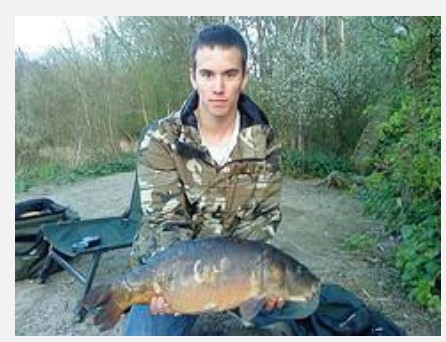

**Imagenet-Choice10 Example**

**Query**: <image>This is an image containing an object. Please identify the species of the object based on the image. The output answer format should be as follows: <think>... </think><answer>species name</answer>
Please strictly follow the format.

Please select the correct species name from the following options: Ursus americanus, shoe shop, brush wolf, essence, malemute, scoreboard, tench, ruddy turnstone, Salamandra salamandra, koala.

**Answer**: tench

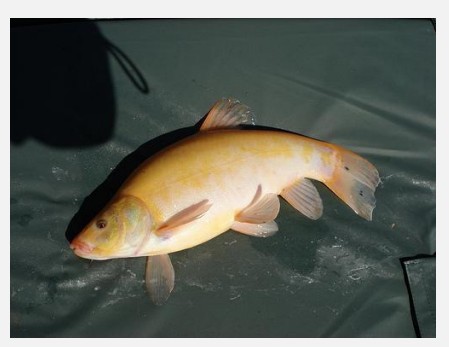

**MMBench Example**

**Query**: <image>Identify the question that Madelyn and Tucker's experiment can best answer.

A. Does Madelyn's snowboard slide down a hill in less time when it has a thin layer of wax or a thick layer of wax?
B. Does Madelyn's snowboard slide down a hill in less time when it has a layer of wax or when it does not have a layer of wax?
C. NaN.
D. NaN.

**Answer:** B

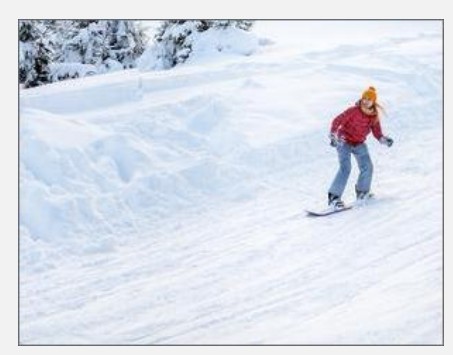

---

**SEED-Bench Example**

**Query**: <image>How many towels are in the image?

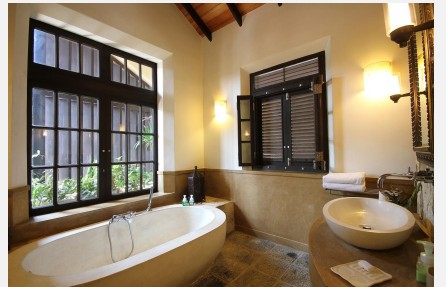

A. One.
B. Two.
C. Three.
D. Four.

**Answer:** A

---

