# OpenReview forum: "SSL4RL: Revisiting Self-supervised Learning as Intrinsic Reward for Visual-Language Reasoning"
_ICML.cc/2026/Conference — ICML 2026 regular_

### Official Review · Reviewer_GezC · 2026-03-06

**Soundness:** 3
**Presentation:** 3
**Significance:** 2
**Originality:** 3
**Overall Recommendation:** 4
**Confidence:** 3

**Summary:**

SSL4RL reframes classic self-supervised tasks (rotation, jigsaw, contrastive, position) as verifiable intrinsic rewards for RL post-training, using GRPO to improve visual grounding and reasoning without human judges. It achieves significant gains on diverse benchmarks, identifies key design factors (task difficulty, model scale, semantic alignment), and generalizes to graph learning.

**Compliance With Llm Reviewing Policy:**

Affirmed.

**Final Justification:**

The author has addressed my concerns. I believe the proposed method is novel and achieves moderate experimental results. Therefore, I am inclined to accept this paper.

**Key Questions For Authors:**

1. It is crucial to carefully compare the image sources of the benchmark samples against the images used for training, and to remove any overlapping images from the training data.
2. The data-scaling experiments evaluate only on MMBench, which is not sufficiently convincing.

**Limitations:**

It may be worth noting that, as the training data scale increases substantially, the performance gains appear limited.

**Strengths And Weaknesses:**

### Strengths:
- Presentation: Well written with clear and smooth logic. The framework is easy to follow, and the SSL task illustrations help readability.
- Originality: The method is novel in turning standard self-supervised objectives into verifiable RL rewards for VLM post-training. It is supported by extensive experiments across multiple settings.
- Soundness: The analysis is fairly comprehensive, with ablations on key factors such as task difficulty (e.g., stronger vs. weaker augmentations / finer rotations), model size (3B vs. 7B), and data volume scaling, which makes the conclusions more convincing.
- Significance:  The paper offers a scalable alternative to human/LLM-judge rewards by turning self-supervised objectives into verifiable RL signals, and shows consistent gains on multiple benchmarks and robustness settings.

### Weaknesses:
- Soundness: Lines 365–368 suggest that the images used for training may come from sources that overlap with the images in the test benchmarks. Even if the questions differ, reusing the same images is still unacceptable, because the model can memorize or develop a deeper familiarity with specific images during training (e.g., recognizing recurring scenes/objects), which undermines the validity of the reported generalization results.
- The data-scaling experiments evaluate only on MMBench, which is not sufficiently convincing.

---

> ### Author Rebuttal · Authors · 2026-03-31
>
> ---
>
> We thank Reviewer GezC for the supportive assessment of our novel framework and clear logic. We appreciate the recognition of our comprehensive experiments and the significance of our scalable alternative to LLM-based rewards. In response to the points raised:
>
> **W1.** Soundness: Lines 365–368 suggest that the images used for training may come from sources that overlap with the images in the test benchmarks. Even if the questions differ, reusing the same images is still unacceptable, because the model can memorize or develop a deeper familiarity with specific images during training (e.g., recognizing recurring scenes/objects), which undermines the validity of the reported generalization results.
>
> **A1.** We thank the reviewer for raising this critical point. We wish to clarify that there is no image overlap between the training and evaluation sets. In the data scaling experiments, we utilize the **MMBench-Test** splits for training (where labels are not publicly available, and we generate verifiable rewards via SSL tasks). For evaluation, we strictly use the **MMBench-Dev** split. The two sets are disjoint in terms of both image-question pairs and the underlying images themselves. We apologize for omitting these spliting details and any resulting confusion. In the revised manuscript, we will include the detailed data split setting.
>
> ---
>
> **W2.** The data-scaling experiments evaluate only on MMBench, which is not sufficiently convincing.
>
> **A2.** We thank the reviewer for this constructive question. To demonstrate the scalability of SSL4RL beyond MMBench, we conduct extensive data-volume scaling experiments on **SEED-Bench**. Following Section 4.5.1, we progressively expand our training volumes as follows: (1) **Base Set**: 4,000 samples sampled from the SEED-Bench pool; (2) **Extended Set**: 8,000 samples by adding 4,000 MMBench samples to the Base Set; (3) **Full Set**: 108,000 samples by adding 100,000 ImageNet samples to the Extended Set. The evaluation is conducted on the remaining portion of SEED-Bench. We select Position as a representative task. As shown in **Table R1**, scaling from the Base Set to the Extended Set yields a significant **+1.81%** improvement (71.58% vs. 69.77%), with further expansion to the Full Set reaching **71.71%**. This monotonic improvement on SEED-Bench verifies the robust scaling effect of our SSL4RL method. We will incorporate these additional experiments into the revised manuscript.
>
> **Table R1. Performance (\%) on the SEED-Bench test set.** IC: Instances Counting, IA: Instance Attributes, SU: Scene Understanding, , IId: Instance Identity, IIn: Instance Interaction, VR: Visual Reasoning, IL: Instance Location, TU: Text Understanding, SR: Spatial Relation.
>
> | Model         | Data Size | IC        | IA        | SU        | IId       | IIn       | VR        | IL        | SR        | TU        | Average   |
> |---------------|-----------|-----------|-----------|-----------|-----------|-----------|-----------|-----------|-----------|-----------|-----------|
> | Qwen2.5-VL-3B | -         | 60.52     | 62.87     | 60.35     | 63.24     | 64.95     | 53.78     | 58.79     | 51.60     | 41.67     | 60.83     |
> | SSL4RL        | 4,000     | 64.20     | 72.51     | 73.56     | 72.75     | 62.89     | 70.69     | 64.62     | 55.25     | 52.38     | 69.77     |
> |               | 8,000     | **65.06** | 75.01     | 75.36     | **74.06** | **67.01** | **74.92** | 65.24     | 56.93     | 57.14     | 71.58     |
> |               | 108,000   | 64.77     | **75.33** | **75.27** | 73.84     | **67.01** | **74.92** | **65.95** | **57.84** | **63.10** | **71.71** |
>
> ---
>
> **Q1.** It is crucial to carefully compare the image sources of the benchmark samples against the images used for training, and to remove any overlapping images from the training data.
>
> **A3.**  Please see **A1**.
>
> ---
>
> **Q2.** The data-scaling experiments evaluate only on MMBench, which is not sufficiently convincing.
>
> **A4.** Please see **A2.**
>
> ----

---

> > ### Author Rebuttal · Reviewer_GezC · 2026-04-01
> >
> > Thank you for your detailed reply. My concern has been resolved.

---

> > > ### Author Response · Authors · 2026-04-04
> > >
> > > We thank the reviewer for the positive feedback and thoughtful questions. In the revised manuscript, we will incorperate the detailed data split setting and the extended data-scaling experiments as promised.

---

### Official Review · Reviewer_4UNt · 2026-03-11

**Soundness:** 2
**Presentation:** 3
**Significance:** 2
**Originality:** 2
**Overall Recommendation:** 4
**Confidence:** 3

**Summary:**

This paper proposes SSL4RL, a framework that turns self-supervised learning tasks into intrinsically verifiable reward signals for reinforcement-learning-based post-training of vision-language models, aiming to reduce their reliance on language priors and improve visual grounding. It instantiates this idea with four pretext tasks and optimizes the resulting rewards with GRPO in a unified corruption–target–reward formulation. Empirically, the method improves performance on vision-language reasoning, ImageNet-style classification, open-ended image captioning, and robustness benchmarks, while also showing that task choice and task difficulty are crucial to downstream gains. The paper further argues that this perspective is not limited to vision by extending the same idea to graph learning and reporting substantial improvements there.

**Compliance With Llm Reviewing Policy:**

Affirmed.

**Final Justification:**

I think the reviewer’s rebuttal has addressed most of my concerns, so I am inclined to accept.

**Key Questions For Authors:**

- Do the authors have a more principled criterion for predicting a priori which SSL rewards will help a given downstream capability, beyond empirical trial-and-error?
- Could the authors provide either human evaluation or an additional evaluation setting?
- The paper argues that SSL4RL is a general framework. Could the authors strengthen this claim with a clearer discussion of what properties a base model must satisfy for SSL4RL to work reliably?

**Limitations:**

Yes

**Strengths And Weaknesses:**

Strengths:
- Well-motivated problem formulation with a clean and compelling perspective shift.
- The experimental section is broad and generally convincing. Authors do not stop at reporting gains, also study which SSL tasks transfer well, how task difficulty affects downstream benefit, how performance scales with more data, and whether multiple rewards combine additively.

Weakness:
- The algorithmic novelty is somewhat limited. While the paper’s framing is elegant, the technical core is still relatively lightweight. Existing SSL pretext tasks are converted into reward functions, and optimization is performed with standard GRPO. As a result, the work is not a fundamentally new modeling method.
- The analysis section gives intuitive explanations and some attention visualizations, but the evidence remains largely qualitative. For example, the claims about reduced language bias, sharper grounding, and anti-commonsense effects are plausible, but the paper does not provide deeper representation analysis to explain how the RL updates induced by different SSL rewards reshape the model.
- The framework does not provide a principled recipe for how to choose, calibrate, or combine rewards in a robust way.
- Additionally, the paper contains an excessive number of dashes, which negatively affects the reading experience.

---

> ### Author Rebuttal · Authors · 2026-03-31
>
> We thank Reviewer 4UNt for the positive feedback of our paper. We now address your concerns as follows.
>
> ---
>
> **W1.** The algorithmic novelty is somewhat limited.
>
> **A1.** We thank the reviewer for appreciating our "elegant framing." While SSL4RL builds on existing SSL tasks and GRPO, its novelty lies in a **paradigm shift** from representation learning to **alignment and reasoning** for VLMs:
> * SSL4RL pioneers to systematically repurpose these objectives as **intrinsically verifiable rewards** for RL-based post-training, offering a scalable alternative to AI-as-a-judge.
> * We demonstrate that tasks often deemed "too simple" for traditional SSL, such as **Rotation**, emerge as effective verifiable anchors for enhancing VQA reasoning.
> * We provide a comprehensive study on SSL4RL—such as the **positive data scaling** and the **non-additivity of rewards**—which offer new, non-trivial insights.
> We believe the framework’s elegance, effectiveness, and generalizability to other domains represent a timely contribution to multimodal alignment.
>
> ---
>
>   **W2.** The evidence remains largely qualitative.
> **A2.**  To quantify visual reliance over language priors, we evaluate on **HallusionBench** [1], which tests overriding linguistic expectations when they conflict with visual input. As shown in **Table R1**, SSL4RL models show consistent gains, notably **+6.10%** for Position. This provides quantitative evidence of reduced language bias and improved visual reliance. We will include these experiments in the revision.
>
> **Table R1. Test accuracy (%) on HallusionBench.**
>
>  | Model | Performance|
> |-|-|
> | Qwen2.5-VL-3B | 46.27 |
>  | Rotation | 47.42 (+1.15) |
> | Jigsaw | 49.32 (+3.05) |
>  | Contrastive| 50.26 (+3.99) |
> | Position| 52.37(+6.10) |
>
> [1]. Guan, Tianrui, et al. "Hallusionbench: an advanced diagnostic suite for entangled language hallucination and visual illusion in large vision-language models." CVPR, 2024.
>
> ---
>
> **W3.** The framework does not provide a principled recipe for how to choose, calibrate, or combine rewards in a robust way.
>
> **A3.**  Please see **A5.**
>
> ---
>
> **W4.** Tthe paper contains an excessive number of dashes.
>
> **A4.** We apologize for the excessive use of dashes and will replace redundant dashes with standard punctuation in the revision.
>
> ---
>
> **Q1.** Do the authors have a more principled criterion for predicting a priori which SSL rewards will help a given downstream capability, beyond empirical trial-and-error?
>
> **A5.** We thank the reviewer for this insightful question. Predicting SSL transferability is an **open challenge** in the broader SSL field. While various predictors have be proposed, there is still no universally accepted metrics.
> - **Beyond Single Tasks.** To avoid selecting a suboptimal task, we experiment with **curriculum task combination**, scheduling tasks by increasing difficulty. This mitigates conflicting reward dynamics from naive mixing. **Results show this curriculum strategy outperforms any single SSL task**, achieving more stable and higher performance.
> - **Future Predictors:** Developing task-agnostic predictors for SSL4RL remains promising. We will add this discussion to the Limitations section.
>
> **Table R2. Combination for SSL4RL tasks. Test on MMBench, 3B-models.**
>
> |   | Test acc (%) |
> |-|-|
> | Best Single Task | 80.38  |
> | Naive Combination| 78.77 |
> | **Curriculum Combination** | **82.14**|
>
> ---
>
>  **Q2.** Could the authors provide either **human evaluation** or an **additional evaluation setting**?
>
> **A6.**  To verify our automated evaluation pipeline, we conducte **human inspection** on MMBench outputs. Two CS PhD students review reasoning paths and final answers. As shown in **Table R3**, the maximum deviation between automated metrics and human judgment is only **0.83%**, confirming that our pipeline serves as a **highly reliable proxy** for model performance. We will include these details in the Appendix.
>
> **Table R3. Comparison between Automated Metrics (Table 1) and Human Evaluation.**
>
> |  | Table 1 | Human Evaluation |
> |-|-|-|
> | Qwen2.5-VL-3B | 72.99 | 72.83 (-0.16) |
> | Rotation | 80.38  | 80.25 (-0.13)  |
> | Jigsaw | 77.82  | 76.99 (-0.83)  |
> | Contrastive| 69.27  | 69.14 (-0.13) |
> | Position| 80.08 | 80.80(+0.72)  |
>
> ---
>
> **Q3.** The paper argues that SSL4RL is a general framework. Could the authors strengthen this claim with a clearer discussion of **what properties a base model must satisfy** for SSL4RL to work reliably?
>
> **A7.**  For SSL4RL to work as a general framework, a base model must satisfy:
> * **Fundamental Perception-Reasoning Alignment.** The model must have basic cross-modal alignment; otherwise SSL tasks become too sparse to guide RL.
> * **Differentiable Capability Gap.** SSL4RL is most effective when a gap exists between latent capacity and task performance, providing a **rich optimization space**. Gains are marginal if the model already masters similar objectives.
>
> We will add this to the "Scope and Limitations" section.
>
> ---

---

> > ### Author Rebuttal · Reviewer_4UNt · 2026-04-04
> >
> > Thanks, the reviewer’s rebuttal has addressed most of my concerns, so I will change the score.

---

> > > ### Author Response · Authors · 2026-04-04
> > >
> > > We are sincerely encouraged that our rebuttal has addressed your primary concerns, and truly appreciate your decision to adjust the score. Following your suggestions, we will incorporate the curriculum combination analysis, the HallusionBench results, the human evaluation, and the principles for model choice into the revised manuscript. Thank you again for your constructive guidance throughout this process.

---

### Official Review · Reviewer_zNr1 · 2026-03-12

**Soundness:** 3
**Presentation:** 3
**Significance:** 2
**Originality:** 2
**Overall Recommendation:** 5
**Confidence:** 3

**Summary:**

They introduce Self-Supervised Learning as a technique to generate verifiable tasks for GRPO on vision. They used four tasks: Rotation Prediction, Jigsaw Puzzles, Contrastive Learning, and Patch Position Prediction. They found out that when they trained models with GRPO using this technique, the model performance on several benchmarks and other tasks like image captioning improves. They also found out that their method has higher robustness to corrupted images. They also tested their method on graph tasks using different graph SSL tasks and showed that their method works in that setting as well.

**Compliance With Llm Reviewing Policy:**

Affirmed.

**Final Justification:**

The rebuttal fully addresses all my concerns.

**Key Questions For Authors:**

1. You mentioned that the task difficulty should match the model's capacity. Is there any experiment that demonstrates this? For example, if the Rotation task works best for the 3B model, and the Jigsaw task (which is harder) works best for the 7B model, then that claim would be supported. I couldn't find such evidence in the paper.
2. Can you add standard deviation bounds to Tables 4 and 6? For me, it is even enough if you evaluate multiple times (using a non-zero temperature, a reasonable value) and report that deviation.

**Limitations:**

Yes

**Strengths And Weaknesses:**

## Strengths

1. The use of SSL for the RL training phase is novel and probably of interest to many.
2. The performance gains are great.
3. The graph experiment shows the technique's usability in other modalities.
4. It doesn't rely on other, stronger MLLMs for training.
5. Their method can use more training data and get better results.

## Weaknesses

1. Task combination: They showed that when they merge their tasks, the model performance drops in comparison to the case where they only use one of the SSL tasks. This is a big weakness, as frontier models usually have to mix many tasks in their RL training phase. I think it is worth investigating whether a bigger model has the capacity to absorb all tasks and gain performance when trained with all of them.
2. The task sensitivity makes training with this technique difficult. Like you cannot choose an SSL task and expect performance improvements. You should test different SSL tasks and choose the one that works.
3. The gains on the 7B model are small, and without standard deviations, it is hard to check if they are just noise or meaningful improvements.

---

> ### Author Rebuttal · Authors · 2026-03-31
>
> We thank Reviewer zNr1 for recognizing the novelty of SSL for verifiable RL training, and for highlighting our performance gains, scalability, and cross-modality usability. We now address the raised weaknesses and questions.
>
> ---
>
> **W1.** Task combination.
>
> **A1.**  We thank the reviewer for this critical observation.
> - **Performance Drop in Naive Mixing.** We assume th performance drop comes from a simpling mixing. The model struggles to find a unified representation that satisfies all constraints, leading to interference rather than synergy.
> - **Solution: Curriculum Combination.** We adopt **curriculum training** with increasing task difficulty, enabling gradual adaptation and avoiding gradient interference. As shown in **Table R1**, this strategy outperforms any single SSL task (82.14% vs. 80.38%) and scales to 7B models (89.26% vs. 87.73%).
>
> In the revised manuscripts, we will add these experiments.
>
> **Table R1. Combination for SSL4RL tasks.**
>
> |Recipe|3B|7B|
> |-|-|-|
> | Best Single Task | 80.38     | 87.73     |
> | Naive Combination          | 78.77     | 85.78     |
> | **Curriculum Combination** | **82.14** | **89.26** |
> ---
>
> **W2.** The task sensitivity makes training with this technique difficult.
>
> **A2.** Predicting SSL transferability is an **open challenge** in the broader SSL field. While various predictors have be proposed, there is still no universally accepted metrics.
> - **Beyond Single Tasks.** In **Q2**, we show the curriculum strategy outperforms any single SSL task. This mitigate the risk of selecting a suboptimal task.
> - **Future Predictors:** Developing task-agnostic predictors for SSL4RL remains promising. We will add this discussion to the Limitations section and provide a preliminary selection guideline based on our empirical observations.
> ---
>
> **W3.** The gains on the 7B model are small, and without standard deviations.
>
> **A3.** Please see **Q2**.
>
> ---
> **Q1.**  Any experiment that demonstrates task-difficulty assumption?
>
> **A4.** To verify the dififuculty-capacity assumption, we design two harder SSL4RL tasks:
> - **Masked Patch Identification:** A harder **Position** task that shifts from spatial prediction to identifying masked details among multiple candidates.
> - **Hard Negative Contrastive:** A more challenging **Contrastive** task that replaces random negatives with semantically similar images retrieved by DINOv2.
> As shown in **Table R2**, harder tasks consistently outperform their base counterparts on 7B models, validating our hypothesis. We apologize for any confusion and will add these details in the revision.
>
> **Table R2. Test acc (%) of harder tasks for 7B models.**
>
> |                       | MMBench   | SEED-Bench | V*        | RealWorldQA | BLINK     | MME       | Averagge  |
> |-|-|-|-|-|-|--|-|
> | Qwen2.5-7B            | 86.37     | 74.70      | 73.29     | 65.88       | 45.50     | 45.59     | 65.22     |
> | Position              | 86.25     | 75.56      | 77.48     | 67.45       | 46.76     | 47.57     | 66.84     |
> | **Mask (Harder)**     | **86.85** | **76.83**  | **78.01** | **68.88**   | **49.28** | **49.03** | **68.14** |
> | Contrastive           | 86.25     | 75.27      | 76.96     | 66.66       | 45.60     | 45.44     | 66.03     |
> | **Negative (Harder)** | **87.78** | **75.79**  | **79.05** | **70.58**   | **52.23** | **48.61** | **69.00** |
> ----
>
> **Q2.** Can you add standard deviation bounds to Tables 4 and 6?
>
> **A5.**  We re-evaluate Tables 4 and 6 using $temperature=0.3$, $top_p=0.9$, and $top_k=50$, with three independent runs per model. As shown in **Tables R3**, while non-zero temperature introduces slight variations, the gains of SSL4RL models remain consistent and statistically significant across benchmarks. Standard deviation results for Table 4 were omitted due to word limits but will be provided as soon as space permits.
>
> **Table R3. Mean and std for Table 6.**
>
> | Category | Model       | MMBench           | SEEDBench         | V*                | RealWorldQA       | BLINK             | MME               | Average   |
> |-|-|-|-|-|-|-|-|-|
> | Base     | Qwen2.5-7B  | 85.99 +- 0.18     | 74.73 +- 0.18     | 73.00 +- 0.42     | 66.01 +- 0.48     | 46.16 +- 0.30     | 45.69 +- 0.15     | 65.26     |
> | SSL4RL   | Rotation    | **87.74 +- 0.21** | 75.05 +- 0.13     | **81.65 +- 0.65** | **68.88 +- 0.26** | **55.61 +- 0.46** | **52.32 +- 0.36** | **70.20** |
> |          | Jigsaw      | 87.53 +- 0.33     | 74.85 +- 0.15     | 80.48 +- 0.74     | 67.66 +- 0.64     | 53.86 +- 0.53     | 47.29 +- 0.13     | 68.61     |
> |          | Contrastive | 86.87 +- 0.36     | 75.18 +- 0.16     | 76.78 +- 0.65     | 66.49 +- 0.67     | 53.03 +- 0.24     | 48.26 +- 0.59     | 67.76     |
> |          | Position    | 87.23 +- 0.14     | **75.47 +- 0.10** | 77.95 +- 0.42     | 67.31 +- 0.28     | 53.10 +- 0.58     | 49.29 +- 0.43     | 68.39     |

---

> > ### Author Rebuttal · Reviewer_zNr1 · 2026-03-31
> >
> > I want to thank the authors for their rebuttal. All of my concerns, with the exception of Q1, have been adequately addressed. As a result, I will raise my score to a Weak Accept.
> >
> > My only remaining concern relates to Q1. Although you demonstrated that a harder version of a task yields better results for a larger model, there is no evidence showing that the easier version works better for a smaller model. Consequently, it remains unclear whether the new task is objectively better or simply a better fit for the model's capacity. However, I recognize this is not a central claim of the paper. If the authors can soften this claim in the next revision or provide more evidence that supports their claim, I would be open to increasing my score further.

---

> > > ### Author Response · Authors · 2026-04-03
> > >
> > > We thank the reviewer for the constructive feedback and for raising the score to Weak Accept. We address the remaining concerns as below.
> > >
> > > ---
> > >
> > > **Evaluation of Harder Tasks on Smaller Models.** We have conducted additional experiments on the **3B base model** to address the remaining concern regarding the “capacity-difficulty” relationship. Following the settings in A4, we evaluated the two harder tasks using a 3B base model. The results in **Table R4** reveal two key insights:
> > > - **Limited gains from too-hard tasks.** For the **Mask** task (harder), the performance is inferior to the **Position** task (58.52% vs. 60.39%). We observed significant reward collision (within the [0.20, 0.30] range) and a lack of convergence during training (see figures in [Anonymous Github](https://anonymous.4open.science/r/Rebuttal_for_SSL4RL/figures_mask_rl_rewards.png)), suggesting that this task may exceed the learning capacity of a 3B model.
> > > - **The original Position is still the best.** While the **Negative** task (harder) outperforms the **Contrastive** task (59.43% vs. 54.11%), it still fails to surpass the **Position** task (60.39%). This is contrastive to our observations with the 7B model.
> > >
> > > These results further verify that a SSL4RL task's effectiveness is coupled with model capacity. We will incorporate these findings (results here and A4) into the revised version and carefully refine our claims to ensure an evidence-based discussion.
> > >
> > > **Table R4. Test performance (%) of harder SSL4RL tasks on 3B models.**
> > > |                       | MMBench   | SEED-Bench | V*        | RealWorldQA | BLINK     | MME       | Average   |
> > > |-----------------------|-----------|------------|-----------|-------------|-----------|-----------|-----------|
> > > | Qwen2.5-3B            | 72.99     | 60.83      | 59.16     | 52.67       | 42.13     | 32.41     | 53.36     |
> > > | Position              | **80.08** | **69.77**  | **68.06** | 59.86       | **46.39** | **38.19** | **60.39** |
> > > | **Mask (Harder)**     | 78.47     | 65.61      | 63.87     | **60.39**   | 44.76     | 38.04     | 58.52     |
> > > | Contrastive           | 69.27     | 61.90      | 60.20     | 58.03       | 45.13     | 30.17     | 54.11     |
> > > | **Negative (Harder)** | **79.60** | **68.14**  | **65.44** | **59.21**   | **46.34** | **37.88** | **59.43** |
> > >
> > > ---
> > >
> > > **Results with std for Table 4 (Extended from A5):** Due to space constraints in our initial response, we present the results across three independent runs in **Table R5**. These results show consistent performance gains for SSL4RL models across subtasks, verifying our previous findings. We will include these complete metrics in the revised manuscript to ensure statistical rigor.
> > >
> > > **Table R5. Mean and std for Table 4.**
> > > | Category | Model       | Logical           | Relation          | Attribute         | Coarse            | Cross             | Single            | Average   |
> > > |----------|-------------|-------------------|-------------------|-------------------|-------------------|-------------------|-------------------|-----------|
> > > | Base     | Qwen2.5-3B  | 53.65 +- 1.11     | 50.52 +- 1.10     | 74.56 +- 1.37     | 66.83 +- 1.25     | 59.41 +- 1.10     | 68.11 +- 0.64     | 62.18     |
> > > | SSL4RL   | Rotation    | **58.91 +- 1.22** | **71.37 +- 1.50** | 78.98 +- 1.19     | 72.39 +- 0.63     | **61.62 +- 0.98** | 73.08 +- 0.94     | **69.39** |
> > > |          | Jigsaw      | 56.22 +- 0.52     | 70.63 +- 1.23     | **79.06 +- 0.33** | **72.90 +- 0.32** | 59.92 +- 1.98     | 72.26 +- 0.35     | 68.49     |
> > > |          | Contrastive | 52.98 +- 1.14     | 70.00 +- 1.07     | 76.78 +- 0.43     | 68.39 +- 1.39     | 54.31 +- 2.70     | 70.92 +- 0.27     | 65.56     |
> > > |          | Position    | 56.68 +- 1.34     | 69.99 +- 2.44     | 77.78 +- 1.31     | 71.38 +- 0.57     | 56.44 +- 1.62     | **73.03 +- 0.91** | 67.55     |
> > >
> > >
> > > ---
> > >
> > > We appreciate your time in reviewing our work and hope these responses resolve your queries. Please let us know if there are any other points you wish to discuss.

---

### Official Review · Reviewer_D42P · 2026-03-13

**Soundness:** 3
**Presentation:** 3
**Significance:** 3
**Originality:** 3
**Overall Recommendation:** 4
**Confidence:** 2

**Summary:**

SSL4RL proposes repurposing SSL pretext task---rotation, jigsaw, contrastive learning, and patch position prediction---as verifiable reward functions for GRPO-based RL post-training of VLMs. The framework eliminates reliance on human preference data or LLM judges by exploiting the intrinsic verifiability of SSL targets. Experiments on Qwen2.5-VL-3B/7B and Gemma3-4B show average improvements of 7.03% on VQA benchmarks at 3B scale, with a Golden-3B oracle comparison demonstrating that SSL4RL closes much of the gap to task-specific golden-reward RL without any downstream labels. An ablation study identifies difficulty–capacity matching, data scaling, and task selection as key design principles. The framework is also extended to graph-domain SSL tasks, yielding notable improvements at 3B scale.

**Compliance With Llm Reviewing Policy:**

Affirmed.

**Final Justification:**

The authors' rebuttal has addressed most of my concerns. I still have some reserves about the generalizability of the curriculum combination and formulation of reward selections. The authors' further experiments on 7B models in the second round are admirable and empirically convincing, but I feel like one needs more theoretical justifications to support these principled qualities of the methods proposed at scale and beyond the selective task domains addressed here. However, overall I do believe this is a solid work and do maintain my recommendation to accept.

**Key Questions For Authors:**

1. Table 6 shows that default Contrastive at 3B produces negative transfer (−3.72% on MMBench), and Contrastive/Position at 7B yield negligible improvement on primary benchmarks. How would the authors recommend practitioners select an SSL task prior to training, and is there a reliable predictor of which task will transfer positively for a given model–benchmark combination?

2. The graph SSL4RL results at 7B (Table 18) show negative or near-zero improvement compared to the base model. Does this reflect the same capacity–difficulty mismatch as the VLM 7B case, and if so, what harder graph SSL tasks were explored or could be proposed?

**Limitations:**

yes

**Strengths And Weaknesses:**

**Strengths**

1. Elegant reward formulation. I think the core idea is really clever, as the SSL tasks provide intrinsically verifiable, label-free reward signals that require no human annotations, external verifiers, or LLM judges. This is a principled and practically attractive alternative to existing RLHF and RLVR pipelines.

2. Golden-3B oracle comparison. Showing that SSL4RL closes most of the gap to task-specific golden-reward RL (e.g., 81.35% vs. 84.93% on MMBench, 69.80% vs. 73.21% on SEED-Bench) is the strongest quantitative argument in the paper and directly addresses the "how competitive is this?" question.

3. Comprehensive ablation study. The difficulty scaling, data volume, model choice, and task combination ablations offer principled design guidance rather than just reporting final numbers. The difficulty–capacity matching principle feels well-supported by the evidence presented.

**Weaknesses**

1. 7B gains are modest and inconsistent on primary benchmarks. Table 6 shows Rotation improves MMBench by only 1.36% and SEED-Bench by 0.86% at 7B. Contrastive and Position at 7B yield essentially no improvement on these primary benchmarks. I think the larger gains on BLINK (+8.83%) and V* (+7.86%) are encouraging, but these are not the paper's main benchmarks. The capacity–difficulty mismatch is acknowledged, but I would consider incorporating the harder task results (currently Appendix J) into the main results rather than leaving the 7B story unresolved in the primary tables.

2. Reasoning vs. perception conflation. The paper frames SSL4RL as inducing "natural language reasoning paths," but response lengths (Figure 9) barely change after RL, remaining short (~100–250 tokens). Since the SSL tasks are perceptual by design, I am not convinced the downstream gains reflect improved reasoning rather than improved perceptual representations---and the paper does not attempt to disentangle them, but since this is not my area I may very well misunderstood---I'd love to hear the authors' thought on this

---

> ### Author Rebuttal · Authors · 2026-03-31
>
> We thank Reviewer D42P for recognizing our "elegant reward formulation" as a "practically attractive alternative" to RLHF. Further clarifications:
>
> **W1** Limited 7B gains; incorporate the harder task results (currently Appendix J) into the main results.
>
> **A1** In the revised version, we will incorporate the results of harder tasks (originally in Appendix J) into the main results section (Section 4.3).
>
> **W2** Reasoning vs. perception conflation.
>
> **A2** Here we clarify why SSL4RL induces natural language reasoning paths.
>
> - Our SSL4RL tasks are not simple perception but require high-order reasoning. For example, the base model’s performance on SSL4RL tasks like Position is near **random guess**, indicating necessity for higher-order reasoning to solve SSL4RL tasks (this is counterintuitive as these tasks are easy for human).
> - Regarding response length, we argue that reasoning length does not consistently correlate with quality [1] (we will provide a comparative case study later due to character limits).
> - Empirically, SSL4RL shows significant improvement on **both perception and reasoning tasks**, e.g. +5.88% in Logical Reasoning, +7.21% in Cross-Instance Perception (Table 1).
>
> We acknowledge that sound reasoning relies on accurate perception. **In the revised version, we'll carefully refine our language to avoid characterizing the gains as solely perceptual or reasoning-based, better reflecting the synergistic improvement of SSL4RL.**
>
> [1]. When More is Less: Understanding Chain-of-Thought Length in LLMs. Wu, et al. ICLR, 2026.
>
> **Q1** Choosing SSL tasks before training?
>
> **A3** Predicting SSL transferability is an open challenge in general SSL [1,2]. While various predictors have be proposed [3,4], there are still no universally accepted metrics. Our empirical findings offer a practical roadmap for SSL4RL:
>
> - Robust Baseline Selection. Based on our experiments, **Rotation** is a reliable starting point for vision-language reasoning (Table 1 & 6), considering its consistent performance across scales.
> - Beyond Single Tasks. To mitigate the risk of selecting a suboptimal task, we conduct new experiments with **curriculum task combination**. Unlike naive mixing in Section 4.5.5, this approach schedules tasks by increasing difficulty (Contrastive → Position/Rotation → Jigsaw), which can reduce reward conflicts of naive combination. **This curriculum strategy outperforms any single SSL task** (82.14% vs. 80.38%, Table R1), achieving greater stability and superior performance.
> - Future Predictors. Developing task-agnostic predictors for SSL4RL remains a promising direction. In the revised version, we will include these discussions in the Limitations section and provide a preliminary selection guideline for practitioners.
>
> **Table R1. Combination for SSL4RL tasks. Test on MMBench, 3B-models**
>
> |Recipe|Test acc (%)|
> |-|-|
> |Best Single Task|80.38|
> |Naive Combination (Figure 5)|78.77|
> |**Curriculum Combination (New)**|**82.14**|
>
> [1]. How Well Do Self-Supervised Models Transfer? Linus Ericsson, et. al. CVPR 2021.
>
> [2]. Same Pre-training Loss, Better Downstream: Implicit Bias Matters for Language Models. Liu Hong, et. al. ICML, 2023.
>
> [3]. LogME: Practical Assessment of Pre-trained Models for Transfer Learning. Kaichao You, et. al. ICML, 2021.
>
> [4]. RankMe: Assessing the downstream performance of pretrained self-supervised representations by their rank. Quentin Garrido, et. al. ICML, 2023.
>
> **Q2** Some graph tasks on 7B → limited improvement; reflects the same capacity–difficulty mismatch as 7B VLM? Harder graph SSL tasks?
>
> **A4**
> - We first clarify a **typographical error** in Table 18: the base model average is **49.48%** (not 57.73%). Now SSL4RL-7B yields a **+1.45% average gain**, aligning with our VLM findings.
> - The marginal gains on some specific tasks indeed reflect the **capacity-difficulty mismatch**. To further challenge 7B models, we propose: 1) **Scaling task difficulty**: *e.g.*, increasing masked keywords for **Attribute** task or expanding candidate pools in **Neighbor**. 2) **Hybrid semantic-structural tasks**: *e.g.,* **Fake Edge Detection**, which requires the model to identify perturbed edges by reconciling node attributes with local topology.
>
> While we are now incorporating harder tasks, a single 7B RL training takes \~2 days, making it difficult to complete full benchmarking shortly. However, we **commit to including these results in the revised manuscript** to further demonstrate the generalizability of SSL4RL.
>
> We appeciate the detailed and thoughtful feedback. We notice that some comments refer to the figure/table numbering in our earlier arXiv preprint. To avoid confusion, all citations in this response also refer to the arxiv version. Our submitted version mainly enriches the experimental evaluation in Arxiv Table 1 (3B) and Arxiv Table 6 (7B), expanding from two benchmarks (MMBench, SEED-Bench) to six comprehensive VQA benchmarks (MMBench, SEED-Bench, V*, RealWorldQA, BLINK, and MME).

---

> > ### Author Rebuttal · Reviewer_D42P · 2026-04-03
> >
> > Many thanks to the authors for their reply. I sincerely appreciate the clarifications and additional experiments. Most of my concerns have been resolved. However, with respect to Q1, I am still not fully convinced that the curriculum task combinations yield a reliable esp. generalizable approach to task selection. I also share Reviewer 4UNt's concern regarding the lack of a principled criterion for choosing, calibrating, or combining rewards in the SSL4RL framework, particularly given the paper's current framing.
> >
> > Nevertheless, I still find this to be exciting work and recognize that addressing such a concern is an open question in the field. I therefore would like to maintain my positive rating. I would also like to note again that this is not my primary area, and I may be overestimating the level of contribution expected for work of this kind, which accounts for my low confidence score, (for the AC:) so please weigh my review accordingly.

---

> > > ### Author Response · Authors · 2026-04-04
> > >
> > > We sincerely thank the reviewer for the constructive feedback and for recognizing the value of our work. We appreciate the opportunity to further clarify the generalizability of our curriculum strategy and the rationale behind our task combination.
> > >
> > > - **On the Generalizability and Reliability of Curriculum Combination.** We understand the concern regarding whether curriculum combinations are generalizable. Our approach follows the **"easy-to-hard" principle**, which has been validated as an effective paradigm for complex RL tasks [1, 2]. To demonstrate the **reliability across scales**, we extended our experiments to **7B base models** **(Table R2)**. The results show that the curriculum combination not only outperforms the best single SSL4RL task (87.73% vs. 89.26%) but also significantly surpasses the naive combination by 3.48%.
> > > - **On the Principled Criterion for Reward Combination.** Regarding the concern about the lack of a principled criterion for combining rewards: we agree that a universal theory for reward calibration is an open question. In this work, we adopted a **stage-wise isolation strategy**. Our ablation study (**Table R3**) reveals that separating the most challenging task (Jigsaw) into a later stage leads to the best stability and accuracy (**82.14%**). Based on this complexity-aware heuristic, we adopt a sequential combination: **Contrastive $\to$ Position $\to$ Rotation $\to$ Jigsaw $\to$ All**.
> > >
> > > **We will reframe the section on task combination** in the revised version to provide a more detailed analysis of the curriculum strategy, incorporating these additional experiments and discussions.
> > >
> > > **Table R2. Performance of SSL4RL combinations on 7B models.**
> > >
> > > |                            | Test Accuracy (%) |
> > > |----------------------------|-------------------|
> > > | Best Single Task           | 87.73             |
> > > | Naive Combination          | 85.78             |
> > > | **Curriculum Combination** | **89.26**         |
> > >
> > > **Table R3. Ablation study on curriculum stage variants.**
> > >
> > > | Stage1      | Stage2              | Stage3   | Stage4 | Stage5 | Accuracy  |
> > > |-------------|---------------------|----------|--------|--------|-----------|
> > > | [All]       | -                   | -        | -      | -      | 78.77     |
> > > | Contrastive | Position            | Rotation | Jigsaw | [All]  | **82.14** |
> > > | Contrastive | Position            | Rotation | [All]  | -      | 81.28     |
> > > | Contrastive | Position & Rotation | Jigsaw   | [All]  | -      | 81.94     |
> > > | Contrastive | Position & Rotation | [All]    | -      | -      | 81.03     |
> > >
> > > [1]. Yuan, Ruifeng, et al. "Vl-cogito: Progressive curriculum reinforcement learning for advanced multimodal reasoning." arXiv preprint arXiv:2507.22607 (2025).
> > >
> > > [2]. Lihong Huang, et al. “From Pixels to Logic: A Perception-Reasoning Decomposition Framework for Open-World Referring Expression Comprehension” AAAI, 2026.
> > >
> > > ---
> > >
> > > We once again thank the reviewer for the encouraging comments. We hope our clarifications and additional results can address your concerns, and we are happy to provide further information if needed.

---

### Decision · Program_Chairs · 2026-04-30

**Decision:**

Accept (regular)

**Comment:**

This paper proposes a new framework SL4RL that leverages self-supervised learning tasks as a source of verifiable rewards for RL-based fine-tuning. The work revisits four pretext tasks and formulates them as reward signals optimized with GRPO. Empirical experiments demonstrate encouraging performance gains across different tasks, including VLM reasoning, ImageNet classification, among others.

The reviewers agree that most of the major concerns have been well addressed in the rebuttal, with only a few minor remaining questions, such as evaluation on harder tasks with smaller models. Overall, this is a solid paper. I recommend acceptance.